Manuscript prepared for Atmos. Chem. Phys.
with version 2014/09/16 7.15 Copernicus papers of the LATEX class copernicus.cls.
Date: 19 December 2016

# Characteristics of Brown Carbon in the urban Po Valley atmosphere

F. Costabile[1], S. Gilardoni[2], F. Barnaba[1], A. Di Ianni[1], L. Di Liberto[1], D. Dionisi[1], M. Manigrasso[3], M. Paglione[2], V. Poluzzi[4], M. Rinaldi[2], M.C. Facchini[2], and G. P. Gobbi[1]

[1]Institute for Atmospheric Sciences and Climate (ISAC), National Research Council (CNR), Rome, Italy
[2]Institute for Atmospheric Sciences and Climate (ISAC), National Research Council (CNR), Bologna, Italy
[3]INAIL, Rome, Italy
[4]ARPA ER, Bologna, Italy

*Correspondence to:* F. Costabile (f.costabile@isac.cnr.it)

**Abstract.** We investigate optical-microphysical-chemical properties of Brown Carbon (BrC) in the urban ambient atmosphere of the Po Valley. In situ ground measurements of aerosol spectral optical properties, $PM_1$ chemical composition (HR-ToF-AMS), and particle size distributions were carried out in Bologna. BrC was identified through its wavelength dependence of light absorption at visible wavelengths ($\lambda$), as indicated by the Absorption Ångström Exponent (AAE). We found that BrC occurs in particles with a narrow monomodal size distribution peaking in the droplet mode, enriched in ammonium nitrate and poor in BC, with a strong dependance on OA-to-BC ratios, and $SSA_{530}$ of 0.98±0.01. We demonstrate that specific complex refractive index values ($k_{530}$=0.017±0.001) are necessary in addition to a proper particle size range to match the large AAEs measured for this BrC ($AAE_{467-660}$=3.2±0.9 with values up to 5.3). In terms of consistency of these findings with literature, this study: (i) provides experimental evidence of the size distribution of BrC associated with the formation of secondary aerosol; (ii) shows that in the lower troposphere AAE increases with increasing OA-to-BC ratios rather than with increasing OA - contributing to sky radiometer retrieval techniques (e.g., AERONET); (iii) extends the dependence of AAE on BC-to-OA ratios previously observed in chamber experiments to ambient aerosol dominated by wood burning emissions. These findings are expected to bear important implications for atmospheric modeling studies and remote sensing observations as regards the parametrization and identification of BrC in the atmosphere.

# 1 Introduction

Aerosol has an important role in the Earth's climate with both direct and indirect effects; in addition, it affects air quality and atmospheric chemistry. At present, our understanding of the light-absorbing aerosol types is incomplete (see reviews by Laskin et al., 2015; Moise et al., 2015) An important absorber of solar radiation in the UV-vis region is atmospheric carbonaceous aerosol (IPCC 2013). In the classification of its components proposed by Pöschl (2003), visible-light-absorbing properties ranged between two extremes. On one side, there is Black Carbon (BC), refractory material that strongly absorbs light over a broad spectral range. On the other side, there is colourless Organic Carbon (OC), non-refractory material, with no absorption or little absorption in the UV-vis spectral range. There is a gradual decrease of thermochemical refractiveness and specific optical absorption going from BC graphite-like structures to non-refractive and colorless OC Laskin et al. (2015). A broad range of coloured organic compounds have recently emerged in the scientific literature for their possible role in the Earth's radiative transfer, therefore on its climate (Laskin et al., 2015; Moise et al., 2015).

The term Brown Carbon (BrC) has emerged to describe this aerosol having an absorption spectrum smoothly increasing from the vis to the near-UV wavelengths, with a strong wavelength dependance of the light absorption coefficient ($\lambda^{-2} - \lambda^{-6}$) (Andreae and Gelencsér, 2006; Moosmüller et al., 2011; Bond et al., 2013; Laskin et al., 2015; Moise et al., 2015). BrC lacks a formal analytical definition (Bond et al., 2013). In this study, we will identify BrC through its high values (2-6) of the visible Absorption Ångström Exponent (AAE), a parameter describing the wavelength ($\lambda$) dependent absorption coefficient ($\sigma_a$) of light by aerosol, written as:

$$AAE(\lambda) = -\frac{dln(\sigma_a)}{dln(\lambda)} \tag{1}$$

What is known about BrC aerosol is that it is organic matter having both primary and secondary sources (Laskin et al., 2015). Primary BrC can be emitted together with BC from low-temperature combustion processes, like wood combustion (Andreae and Gelencsér, 2006). Secondary organic aerosol (SOA) formed in the atmosphere also contributes to light-absorbing-carbon (Moise et al., 2015, and references therein), but only a few field measurement studies have analysed the BrC associated to SOA (Zhang et al., 2011; Saleh et al., 2013; Zhang et al., 2013; Gilardoni et al., 2016).

Numerous evidences indicate increased absorption towards UV for aerosol particles high in nitrate (e.g., Jacobson, 1999; Zhang et al., 2013), sulfate (Lee et al., 2013; Song et al., 2013; Powelson et al., 2014; Lin et al., 2014), and ammonium (Shapiro et al., 2009; Noziere et al., 2009; Bones et al., 2010; Noziere et al., 2010; Sareen et al., 2010; Updyke et al., 2012; Flores et al., 2014; Lin et al., 2015). Lin et al. (2014) reported the formation of light-absorbing SOA constituents from reactive uptake of isoprene epoxydiols onto preexisting acidified sulfate seed aerosol as a potential source of secondary BrC under tropospheric conditions. Powelson et al. (2014) discussed BrC formation by aqueous-

phase carbonyl compound reactions with amines and ammonium sulfate. Lee et al. (2013) studied the likely but uncertain effect of sulfate on the formation of light-absorbing materials and organo-

nitrogen via aqueous glyoxal chemistry in aerosol particles. Song et al. (2013) observed significant light absorption at 355 and 405 nm for SOA formed from an $\alpha$-pinene + $O_3$ + $NO_3$ system only in the presence of highly acidic sulfate seed aerosols under dry conditions. Several studies demonstrated the importance of ammonium, both as a catalyst and as a reactant, in the formation of light-absorbing products (Powelson et al., 2014; Laskin et al., 2015). SOA formation can occur in both the gas and

condensed phases. Recently, efficient SOA production has been recognised in cloud/fog drops and water containing aerosol: water soluble products of gas-phase photochemical reactions may dissolve into an aerosol aqueous phase and form SOA through further oxidation, this SOA being referred to as "aqSOA" (Ervens et al., 2011; Laskin et al., 2015). AqSOA formation impacts total SOA mass and aerosol size distributions by adding mass to the so-called "droplet mode" (Ervens et al., 2011). Meng

and Seinfield (1994) showed that the aerosol "droplet mode" in urban areas is the result of activation of smaller particles to form fog, followed by aqueous-phase chemistry and fog evaporation. It was demonstrated that aqSOA formation can affect aerosol optical properties by adding light-absorbing organic material at UV wavelengths (Shapiro et al., 2009; Ervens et al., 2011; Gilardoni et al., 2016). Recently, Gilardoni et al. (2016) demonstrated that in the ambient atmosphere the aqSOA from

biomass burning contributes to the BrC budget and exhibits light absorption wavelength dependence close to the upper bound of the values observed in laboratory experiments for fresh and processed biomass-burning emissions.

Despite the efforts made, relations between optical properties and chemical composition of organic compounds with spectrally variable light absorption (high AAE) are poorly understood (Laskin

et al., 2015). A number of previous works (Shinozuka et al., 2009; Russell et al., 2010; Arola et al., 2011) studied how the organic aerosol (OA) mass fraction ($f_{OA}$) relates to AAE and to Single Scattering Albedo (SSA), the ratio of scattering to extinction, a key parameter in understanding aerosol warming or cooling effect. Results from in situ measurements on the C-130 aircraft (Central Mexico) during MILAGRO (Russell et al., 2010) showed that both organics and dust increase AAE values.

Russell et al. (2010) showed a direct correlation between AAE and $f_{OA}$. On the basis of the same data, Shinozuka et al. (2009) showed that AAE generally increases as $f_{OA}$ or SSA increases. Saleh et al. (2014) burnt a selection of biomasses in a combustion chamber, varying the combustion parameters to obtain a range of BC-to-OA ratios. This ratio, the relative proportions of BC and OA mass, depends on fire characteristics and plume age, and determines aerosol colour from black to

brown to white as the ratio decreases. Saleh et al. (2014) findings link the extent of absorbance to the BC-to-OA ratio for aged and fresh biomass burning aerosols. If confirmed, this link has the potential to be a strong predictive tool for light-absorbing properties of biomass burning aerosols (Bellouin, 2014; Moise et al., 2015). Following the approach of Saleh et al. (2014), Lu et al. (2015) reviewed available emission measurements of biomass burning and biofuel combustion, and found

similar results indicating that AAE of the bulk OA decreases with the increasing BC-to-OA ratio. They conclude that the absorptive properties of OA from biomass/biofuel burning depend strongly on burning conditions and weakly on fuel types and atmospheric processing.

In this study, we investigate optical-microphysical-chemical properties of BrC in the ambient urban atmosphere. In situ ground ambient data of chemical (HR-ToF-AMS), optical (3-$\lambda$ nephelometer

and 3-$\lambda$ Particle Soot Absorption Photometer, PSAP), and microphysical (SMPS and APS) aerosol properties were taken during two field measurements in Bologna, Po Valley. BrC is identified through the AAE of the non-dust bulk aerosol. First (Sect.4.1), we investigate BrC properties by relating the AAE to the aerosol size, and in particular to the aerosol types with known size distribution - e.g., the droplet mode. We match AAE patterns measured for BrC to those theoretically expected for BrC in

the ambient aerosol (based on the Mie theory). The AAE and aerosol size are then related to PM$_1$ major chemical components (nitrate, OA, BC, sulfate, and ammonium), and to the BC-to-OA ratio. Then, we show a case study to illustrate the major features of BrC (Sect.4.3). Finally, findings are discussed in comparison with the literature to explore their general validity (Sect.4.4).

## 2   Experimental

Optical, chemical, and microphysical aerosol properties were measured, in the framework of the Supersite project funded by Emilia Romagna region, at the urban background site of Bologna (44 ° 31' 29" lat, 11° 20' 27' lon), in the Po Valley (Italy). Two measurement campaigns lasting one month were carried out: October 22 - November 13, 2012 (Fall campaign), and February 1-27, 2013 (Winter campaign). Measurements set-up is described below.

### 2.1   Measurement cabins and sampling lines

Equipment was set up in two different cabins, located side by side. Optical properties and coarse fraction size distributions were measured in the same cabin, all the instruments set up on the same inlet system equipped with a PM$_{10}$ head. In the cabin, external air was pumped into a stainless steel tube (length = 4.0 m) by an external pump ensuring a laminar flow (Reynolds number <2000). The

cabin was kept at a temperature of 20-25°C. The difference between air temperature and dew point was enough to dry the sampled air. Chemical properties and fine and ultrafine particle number size distribution were measured through a separate stainless steel inlet tube equipped with a PM$_1$ head.

### 2.2   Optical Measurements

Spectral optical properties in the visible range were measured online with 5-minute time resolu-

tion. Dry aerosol absorption coefficients, $\sigma_a(\lambda)$, at three wavelengths ($\lambda$ = 467, 530, 660 nm) were measured by a 3-wavelength PSAP (Radiance Research), together with dry aerosol scattering coefficients ($\sigma_s(\lambda)$) at 450, 525, and 635 nm, measured by an integrating nephelometer (Ecotech, mod.

Aurora 3000). Like all filter-based methods, PSAP suffers from a number of measurement artifacts, including an overestimate of absorption due to light scattering effects, and a dependence of measurements on filter transmittance (Tr) Lack et al. (2008); Virkkula (2010); Bond et al. (2013); Backman et al. (2014). We corrected raw PSAP data after the iterative procedure described by Virkkula (2010), where only data with Tr >0.7 were retained. The wavelength-resolved $\sigma_s(\lambda)$ necessary to correct PSAP raw data were taken from nephelometer data corrected for truncation (Anderson and Ogren (1998), Bond (2001), and Müller et al., 2011). The scattering error after the truncation error correction is $\frac{\delta(\sigma_s)}{\sigma_s} = 0.02$ (Bond et al., 2013). The uncertainty of $\sigma_a(\lambda)$ derived from PSAP data after these corrections has been estimated to be $\frac{\delta(\sigma_a)}{\sigma_a} = 0.2$ (Virkkula, 2010; Lack et al., 2008; Bond et al., 1999; Virkkula, 2010; Cappa et al., 2008). The PSAP-derived $\sigma_a(\lambda)$ can be considered an upper limit of the "true" value (Subramanian et al., 2007; Lack et al., 2008).

After all corrections, data were checked (by visual inspection) to find any outlier/low values that could significantly influence data statistics. These values could be due to variability in the measurements or to experimental errors. According to manufacturers: (i) PSAP sensitivity is <1 Mm$^{-1}$, and measurement range is 0-50 Mm$^{-1}$; (ii) the lower detectable limit of the nephelometer is 0.3 Mm$^{-1}$, with calibration tolerance of $\pm$ 4 Mm$^{-1}$, and measurement range 0-2000 Mm$^{-1}$. A few data (124 records having $\sigma_a$ <1 Mm$^{-1}$, less than 20 records with $\sigma_s$ <10 Mm$^{-1}$, and some points with $\sigma_s$ >700 Mm$^{-1}$) were discarded, as they were considered dubious values (comparing to data variability during the field campaigns, illustrated in Figure 1 of the Supplementary material).

## 2.3 Chemical Measurements

Chemical composition of atmospheric aerosol particles were characterized online with a High Resolution Time of Flight Aerosol Mass Spectrometer (HR-ToF-AMS, Aerodyne Research Inc., Billerica) (DeCarlo et al., 2006). The HR-ToF-AMS measured the chemical composition of non-refractory PM$_1$ (nr-PM$_1$), i.e sulfate, nitrate, ammonium, chloride, and organic aerosol. The instrument alternated acquisition in V mode (higher sensitivity and lower mass spectral resolution), and W mode (lower sensitivity and higher mass spectral resolution) every 2.5 minutes. Quantitative information discussed here corresponds to the data collected in V mode. While operating in V mode, the instrument measures particle size distribution based on time of flight (Jimenez et al., 2003). HR-ToF-AMS data were analyzed using SQUIRRELL v1.51 and PIKA v1.10 software (D. Sueper, University of Colorado, Boulder, CO, USA) within Igor Pro 6.2.1 (WaveMetrics, Lake Oswego, OR). Collection efficiency was calculated according to Middlebrook et al. (2012) based on aerosol chemical composition and relative humidity. Data validation was performed by comparison with offline measurements of sulfate, ammonium, and nitrate concentrations in PM$_1$ aerosol samples. The HR-ToF-AMS aerosol sample line was dried below 40% RH with a Nafion drier. The uncertainty of the AMS-derived OA was assumed to be $\frac{\delta(OA)}{OA} = 0.2$ according to Quinn (2008).

## 2.4 Particle Number Size distributions

Particle number size distributions (PNSDs) were measured by combining a commercial Scanning Mobility Particle Sizer (SMPS, TSI mod. 3080 with Long-DMA, TSI mod. 3081, equipped with a water-based Condensation Particle Counter, CPC, TSI mod. 3787), and a commercial Aerodynamic Particle Sizer (APS, TSI). Particles from 14 nm to 750 nm of mobility diameter were sized and counted by the SMPS; particles from 0.5 to 20 $\mu$m of aerodynamic diameter were sized and counted by the APS (the procedure to fit the two PNSDs is described in Sect.3.2). SMPS data were corrected for penetration errors through the sampling line, penetration efficiency due to diffusion losses (calculated according to Hinds (1999)) being higher than 98% for particles bigger than 14 nm. An impactor (50% cutoff diameter = 0.677 $\mu$m) was used to remove larger particles from the SMPS sampling line.

## 3 Data analysis

Data measured by all the instruments were merged in a single 5-minute time resolution dataset. This dataset includes the time series of the following variables: $\sigma_a(\lambda)$ and $\sigma_s(\lambda)$, OA, $NO_3^-$, $SO_4^{2-}$, $NH_4^+$, and PNSD. Raw data were subjected to various "cleaning" processes as described in Sect.2, and then analysed as described in this section. The time series subjected to data analysis includes 11910 time points covering 40 days of measurements (5317 time points in the Fall and 5650 time points in the Winter). This time series includes missing values of some variables, related to instrument failures or data filtering as described above. The length of the complete time series (i.e., with no missing value) varies from variable to variable (10897 time points for OA, 10361 time points for $NO_3^-$, 8999 time points for $SO_4^{2-}$, 9677 time points for $NH_4^+$, 2656 time points for the PNSD, 2367 time points for AAE, SSA, and BC, 1820 time points for $f_{BC}$, $f_{OA}$, $f_{NO3}$, $f_{SO4}$, $f_{NH4}$ and OA-to-BC). Overall, about 1500 data points have the complete set of coincident measurements of all the variables addressed.

## 3.1 Inference of the optical Black Carbon mass concentration

The wavelength ($\lambda$) dependent BC absorption coefficient ($\sigma_{aBC}(\lambda)$), and equivalent BC mass concentration, were calculated using the $AAE_{BC}$ attribution method. The measured absorption coefficient at 660 nm ($\sigma_a(660)$) was used to derive $\sigma_{aBC}(530)$, and then the BC mass concentration, assuming: (i) BC is the only light absorbing species at 600 nm, (ii) a known value of $AAE_{BC}$ at 530-660 nm (see below), and (iii) a BC Mass Absorption Efficiency at 530 nm of 10 m$^2$ g$^{-1}$ (as indicated by PSAP manufacturer). In literature, $AAE_{BC}$=1 is a commonly used value for both externally mixed BC and internally mixed BC. In fact, $AAE_{BC}$ for externally mixed BC was predicted to be 1 for particles with diameter < 50 nm (e.g., Bergstrom et al., 2002; Moosmüller et al., 2011), but can range from 0.8 to 1.1 for particle diameters of 50-200 nm (Gyawali et al., 2009). For ambi-

ent particles, which can be internally or externally mixed, $AAE_{BC}$ at visible wavelengths has often been observed to be larger than 1 (Lack and Langridge, 2013; Shinozuka et al., 2009, and references therein). Theoretical calculations have shown that the $AAE_{BC}$ for internally mixed BC can vary

from 0.55 (e.g., Bahadur et al., 2012) to an upper limit of $\sim 1.7$ (e.g., Lack et al., 2008) depending on particle size, coating, core and wavelength. In Figure 2 of the Supplementary material we show numerically simulated (Mie Theory) values of $AAE(d_p, \lambda, m)$ resolved by particle diameter ($d_p$) and complex refractive index ($m(\lambda)$) at visible wavelengths ($\lambda$) for BC (Sect.3.4). It is shown that the AAE of BC tends to 1 for the smaller BC particles only, and can differ significantly from 1 for the

larger BC particles. Based on these results and on previous works, we decided to use $AAE_{BC}$=1.1. The uncertainty $\delta(AAE_{BC})$ was set to 22% ($\frac{\delta(AAE_{BC})}{AAE_{BC}}$=0.22) according to Lack and Langridge (2013). BC uncertainty ($\delta(BC)$) was derived propagating this $\delta(AAE_{BC})$ together with the uncertainty of PSAP-derived $\sigma_a$ (see Sect.2.2).

We discarded data possibly affected by desert dust (43 records over 5 days of measurements) to

205 ensure that the equivalent BC mass concentration is not affected by desert dust contamination (assumption i above). Dust-free aerosol conditions were identified based on the analysis of aerosol spectral optical properties, increasingly applied to gather information on aerosol type (e.g., Bergstrom et al., 2002; Shinozuka et al., 2009; Russell et al., 2010; Arola et al., 2011; Costabile et al., 2013). In particular, we followed the methodology proposed by Costabile et al. (2013) which identifies the

210 aerosol dominated by dust by a distinctive combination of scattering and absorption Ångström Exponents (SAE, AAE) and Single Scattering Albedo (SSA) spectral variation (dSSA). Data points of the time series fulfilling this distinctive combination (indicated in Table 3 of the Costabile et al. (2013)'s paper) were excluded from the analysis.

### 3.2 Fitting procedure for the particle number size distribution

Particle number size distributions (PNSDs) were measured by two different instruments (SMPS and APS, Sect.2). These data were merged to obtain one PNSD based on particle electrical mobility diameters ($d_m$) ranging from 14 nm to 14 $\mu$m. PNSDs measured by APS are based on aerodynamic diameters ($d_a$); these data were converted to PNSDs based on $d_m$ according to Eq.2 (Khlystov et al., 2004; Seinfeld and Pandis, 2006):

$$220 \quad d_m = \chi \frac{C_c(d_m)}{C_c(d_a)} \frac{d_a}{(\frac{\rho_p}{\rho_0 \chi})^{1/2}} \tag{2}$$

where $\chi$ is the shape factor, $C_c(d_m)$ and $C_c(d_a)$ are the slip correction factors based on $d_m$ and $d_a$ respectively, $\rho_p$ is the particle density, and $\rho_0$ is the unit density (1 g$\cdot cm^{-3}$). In applying Eq.2 to convert APS data, we assumed: $d_m$ represents the true particle diameter; $C_c(d_m) = 1$ and $C_c(d_a) = 1$ (continuum regime); $\chi = 1$ (spherical particles); $\rho_p$ continuously varying from 1.6 to 2 g$\cdot cm^{-3}$.

PNSDs (i.e., $n_N(log_{10}d_m) = \frac{dN}{dlog_{10}(d_m)}$) measured by SMPS and APS (PNSD$_{SMPS}$, PNSD$_{APS}$) overlap for d$_m$ ranging from 460 nm to 593 nm. In this size range, PNSD$_{SMPS}$ and PNSD$_{APS}$ were replaced by PNSD$_{fitted}$. PNSD$_{fitted}$ was assumed to vary according to a power-law function (Junge size distribution) (Khlystov et al., 2004; Seinfeld and Pandis, 2006) (Eq.3):

$$n_N(log_{10}d_m) = \frac{c}{d_m^\alpha},$$  (3)

The coefficients c and $\alpha$ were calculated by an iterative procedure: (i) c was randomly initialized from 0 to 1000; (ii) $\alpha$ was calculated by Eq.3 constraining values from 2 to 5, as typically found for atmospheric aerosols (Seinfeld and Pandis, 2006). PNSD$_{fitted}$ replaced PNSD$_{APS}$ and PNSD$_{SMPS}$ when their relative difference ($\delta(PNSD)$, Eq.4):

$$\delta(PNSD) = \frac{|PNSD_{SMPS} - PNSD_{APS}|}{max[PNSD_{SMPS}, PNSD_{APS}]}$$  (4)

was larger than 0.1 cm$^{-3}$. This procedure was considered acceptable if: (i) the minimum mean squared error between PNSD$_{fitted}$ and PNSD$_{APS}$ was less than 1%; (ii) correlation coefficients between PNSD$_{fitted}$ and PNSD$_{SMPS}$, and between PNSD$_{fitted}$ and PNSD$_{APS}$ were larger than 0.8 (98 of the records did not verify these conditions, and were checked by visual inspection: 94 of them were discarded, and 4 accepted). The final dataset contained PNSD data based on d$_m$ from 14.1 to 429.4 nm measured by the SMPS, from 446.1 to 699 nm generated by the fitting procedure, and from 0.7 to 14 $\mu$m measured by the APS . The Particle Surface Size Distribution (PSSD, i.e. $n_S(log_{10}d_m) = \frac{dS}{dlog_{10}(d_m)}$) and Particle Volume Size Distribution (PVSD, i.e. $n_V(log_{10}d_m) = \frac{dV}{dlog_{10}(d_m)}$) were calculated from this PNSD under the hypotheses of spherical particles (Hinds, 1999; Seinfeld and Pandis, 2006).

## 3.3 Principal component analysis of PNSD

PNSDs were statistically analysed through Principal Component Analysis (PCA) to identify key aerosol types with known modality. The relevant methodology was described in a previous study by Costabile et al. (2009). In short, Principal Components (PCs) retained in the analysis were arranged in decreasing order of variance explained ($\varkappa_k$, called eigenvalue of PC$_k$), PC1 being the component explaining the largest $\varkappa_k$. The $k^{th}$ eigenvector is composed of scalar coefficients describing the new PC$_k$ as a linear combination of the original variables (the original variables are the time series of dN/dlog(d$_p$)). Factor loadings of PC$_k$ represent the relative weight of the original variables in the PC$_k$ re-scaled. Loadings thus show the "mode" of the PNSD associated to the PCs. Factor scores of PC$_k$ represent the time series of PC$_k$ values in the new coordinates of the space defined by the PCs. Scores thus represent the PC$_k$ values in the time series of the original variables.

PCA retained three principal components (PC1-PC3) explaining approximately 80% of the variance. Factor loadings and diurnal cycles of scores for PC1-PC3 are illustrated in Figure 3 of the

supplementary material, while Pearson's correlation coefficients r between these PCs and the other variables measured are shown in Table 1. Table 1 and Table 2 of the supplementary material show relevant r values in the Winter and in the Fall, and relevant Bonferroni adjusted probabilities (p values), respectively. These PCs were interpreted as follows:

PC1 is the largest component in terms of variance explained (51 %). Loadings peak in the 80-300 nm size range. Factor scores correlate to OA and BC. Weekly diurnal cycles of these scores are higher on working days and in the road-traffic rush hour. This PC represents the aerosol enriched in OA originating in the traffic rush hour in the urban area, due to local emissions (e.g., Costabile et al., 2009; Brines et al., 2015, and references therein).

PC2 explains 14 % of the variance. Loadings peak in the ultrafine size range (approx. 100 nm). Factor scores show diurnal cycles higher at night and in the Winter, with a slightly larger contribution during the week-ends. It correlates (inversely) to aerosol size and (directly) to $f_{OA}$; in the Winter it correlates to both $f_{OA}$ and $f_{BC}$ (0.40, p<0.001, Tab.1 of the Supplementary material). This PC should represent the nocturnal urban aerosol related to residential heating emissions.

PC3 explains 13% of the variance. Loadings peak in the larger accumulation mode size range (from 0.3 to 1 $\mu$m). Diurnal cycles of scores are higher in daytime. It is inversely correlated to $f_{BC}$, and directly to nitrate, sulfate and ammonium. This PC represents the droplet mode aerosol poor in BC, previously found to originate in fog droplets, cloud droplets, and wet aerosol particles, due to aqueous phase processing (John, 1990; Meng and Seinfield, 1994; Seinfeld and Pandis, 2006; Ervens et al., 2011).

### 3.4 Numerical simulations of AAE

Values of AAE($d_p, \lambda, m$) resolved by diameter $d_p$, radiation wavelength ($\lambda$), and complex refractive index ($m_{(\lambda)} = n_{(\lambda)} - ik_{(\lambda)}$), were numerically simulated according to Mie theory (e.g., Bohren and Huffman, 1983; Moosmüller et al., 2011). The aim was to reproduce patterns expected for BrC, BC and the urban background aerosol impacted by biomass burning emissions, as these were abundant in the study area. Simulations are illustrated in Fig. 2 of the supplementary material, the relevant methodology being described in a previous study by Costabile et al. (2013).

To simulate patterns expected for BrC in the urban ambient aerosol, we used $\lambda$ dependent complex values of $m_{(\lambda)}$ inferred during CAPMEX for an air mass with AAE$_{405-532}$=3.8 (standard deviation=3.4), characterised by high Organic Carbon (OC) to sulfate ($SO_4^{2-}$) ratio and high nitrate ($NO_3^-$) to $SO_4^{2-}$ ratio (Flowers et al., 2010; Moise et al., 2015). These are $m_{467}$=1.492-0.026$i$, $m_{530}$=1.492-0.017$i$, $m_{660}$=1.492-0.014$i$, the uncertainty for $n_{(\lambda)}$ and $k_{(\lambda)}$ being set to $\pm$0.01 and $\pm$0.001, respectively.

To simulate patterns expected for BC we used $\lambda$ independent complex values of $m$ estimated by Alexander et al. (2008) for soot carbon particles at 550 nm: $n$=1.95-0.79$i$ at $\lambda$=467,530,660 nm.

Note in Fig. 2 of the supplementary material the resulting variability with $d_p$ of $AAE_{467-660}$ for BC: values of AAE=1 are only obtained for $d_p << 100$ nm.

To simulate patterns expected for the urban background aerosol impacted by biomass burning emissions we used values of $m_{(\lambda)}$ inferred in a previous study for the smaller accumulation mode particles enriched in BC from biomass burning smoke (Costabile et al., 2013): $m_{467}$=1.512-0.027i, $m_{530}$=1.510-0.021i, $m_{660}$=1.511-0.022i , the uncertainty for $n_{(\lambda)}$ and $k_{(\lambda)}$ being set to $\pm 0.01$ and $\pm 0.001$, respectively.

## 4  Results and discussion

In this section we first identify brown carbon (BrC) and characterize its optical-microphysical-chemical properties (Sect.4.1), then illustrate a case study (Sect.4.2), and finally discuss the findings in comparison with the literature (Sect.4.3).

### 4.1  Brown carbon: identification and features

Several literature studies identify BrC based on the high AAE values in the bulk aerosol, i.e. from 2 to 6 (e.g., Andreae and Gelencsér, 2006; Bond et al., 2013). At a certain range of wavelengths ($\lambda$), these high AAE values depend on several factors, including aerosol size, chemical composition, and aerosol mixing state. First, we analysed the dependence of AAE on aerosol size. We used two different approaches. We calculated the median mobility diameter of the particle surface size distribution (PSSD, Sect.3.2) - $d_{med(S)}$ - to obtain the optically relevant aerosol size representative of the entire particle population. We then analysed the particle number size distribution (PNSD) to find major aerosol types (i.e., Principal Components, PCs, Sect.3.3). We found two components (PC1 and PC2) related to smaller particles and originating from local emissions (road-traffic and residential heating, respectively), and one component (PC3) related to larger particles (droplet mode) originating from the aerosol processing. Second, we analysed the relation between AAE, these PCs, and $d_{med(S)}$, and related these to $PM_1$ major constituents. Relevant statistically significant Pearson's correlation coefficients (r) are shown in Tab. 1, while r values observed in the Winter and in the Fall and the matrix of Bonferroni Probabilities (p) associated to these r values are shown in Tab.1 and Tab.2 of the Supplementary material.

The AAE correlates well with the $d_{med(S)}$ of the aerosol population (r=0.60, p<0.001). In Figure 1 we analyse this relation by comparing field measurements to numerical simulations results (these simulations are based on the Mie theory and are described in detail in Sect.3.4). Measurements show that the AAE increases with the increase of the $d_{med(S)}$ (grey markers). We show the AAE and $d_{med(S)}$ representative of the three different aerosol types identified (PC1, PC2, and PC3). To interpret these measurements we add patterns theoretically expected (through numerical simulations) for three aerosol types: BrC in the ambient aerosol (brown line), a pure BC particle (black line), and

an urban background aerosol impacted by wood burning emissions (grey line). The lowest AAEs measured in the present study tend to values expected for pure BC (black line), but are similar to values calculated for the urban background aerosol impacted by wood burning emissions (grey line) -

both aerosol types which were related to local emissions (PC1 and PC2) match these patterns. Larger AAEs (3.2±0.9) correspond to the droplet mode aerosol (PC3), and are similar to the AAE expected for Brown Carbon (brown line). Note that Fig.1 suggests that the threshold value of $AAE_{467-660}$ distinguishing BrC is AAE>2.3.

The dotted black line in Fig.1 represents the best fit to all the datapoints of the measurements.

This line shows the increase of the AAE with the increase of the $d_{med(S)}$ in the measurements. To interpret this pattern, we note that the brown and grey lines (referring to the simulations of BrC and of the wood burning related aerosol, respectively) show the theoretical increase of the AAE with increasing only the aerosol size - i.e. when $m_{(\lambda)}$ is constant. Comparing these three lines, it is evident that specific values of $m_{(\lambda)}$ are necessary in addition to a proper particle size range to match the large

AAE measured for the droplet mode BrC aerosol. These values are $k_{(467)} = 0.026\pm0.001$, $k_{(530)} = 0.017\pm0.001$, $k_{(660)} = 0.014\pm0.001$, and $n_{467}=1.47\pm0.01$ (Sect.3.4). These are $\lambda$ dependent values consistent with BrC in the ambient atmosphere (Moise et al., 2015) in an air mass having high OC to $SO_4^{2-}$ ratio and $NO_3^-$ to $SO_4^{2-}$ ratio (Flowers et al., 2010). This finding emphasizes that BrC properties depend on both aerosol size distribution and chemical composition (i.e., $m_{(\lambda)}$).

Figure 2 investigates the link between chemical, microphysical, and optical properties. AAE is plotted against $PM_1$ major chemical components (OA, BC, nitrate, sulfates, and ammonium $PM_1$ mass fractions, $f_x$). Grey markers show the longest available time series for AAE and $f_x$, while coloured markers show the datapoints for which the droplet mode aerosol scores (marker colour) and the $d_{med(S)}$ (markers size) were available (see Sect.3). The BrC aerosol population (identi-

fied by AAE>2.3) shows higher $f_{NO3}$ and $f_{NH4}$ and lower $f_{BC}$ values coupled to larger $d_{med(S)}$ and high droplet mode aerosol scores. Relevant average values are as follows (mean±standard deviation): AAE=3.2±0.9, $f_{NO3}$=0.38±0.05, $f_{BC}$=0.01±0.01, $f_{OA}$=0.35±0.04, $f_{SO4}$=0.1±0.02, $f_{NH4}$=0.14±0.01, and $SSA_{530}$=0.98±0.01 (and $\sigma_{a467} = 7.6\pm3.33\ Mm^{-1}$, $\sigma_{s467} = 312\pm64\ Mm^{-1}$, and SAE=0.5±0.4). The relation between AAE and SSA will be further analysed in Figure 5 (in

Figure 4 of the supplementary material we show the relation between the droplet mode aerosol and SSA).

Both Fig.2 and Tab.1 show no direct correlation between AAE and $f_{OA}$. This may be explained by the fact that AAE correlates with larger particles (larger $d_{med(S)}$) of the droplet mode (larger PC3 scores), while the $f_{OA}$ correlates with smaller particles (lower $d_{med(S)}$) from residential heating

emissions (larger PC2 scores). There is instead a significant correlation between AAE and the ratio of OA to BC (r=0.78, p<0.001), a variable indicating either combustion characteristics (higher for biofuels than for fossil fuel combustion) or aerosol ageing (lower for fresh aerosols) (Saleh et al., 2014; Bond et al., 2013). The increase of the AAE with the decrease of the BC-to-OA ratios is

illustrated in Figure 3. Light-grey markers show the longest available time series for AAE and BC-to-OA, while coloured markers correspond to datapoints for which the droplet mode aerosol scores and $f_{NO3}$ were available. The relation observed between AAE and BC-to-OA is an average aerosol property (i.e., for both BC and OA) as suggested by the parametrization of AAE as a function of the BC-to-OA ratio (inner panel of Fig.3) consistent with Lu et al. (2015) (see Sect.4.3). The grey line (referring to all the data points) shows the parametrization for the bulk aerosol. The black line (referring to the subset of all the data with PC3 scores > 0) shows the parametrization for BrC. We note a larger dependence of AAE on the BC-to-OA ratio for BrC (r=-0.86, p<0.001) - as opposed to the bulk aerosol (r=-0.7, p<0.001). However, this can be due to the fact that at low BC-to-OA ratios OA has more weight in dictating AAE.

In Fig. 3 AAE plateaus at values higher than 1 for large BC-to-OA ratios. We acknowledge that BC and AAE derived from PSAP measurements are affected by uncertainty related to the assumption employed to convert raw PSAP measurements to absorption coefficients, which is about $\pm$ 40% based on literature studies (Lack et al., 2008; Nakayama et al., 2010; Lack and Langridge, 2013; Bond et al., 2013; Backman et al., 2014) . This uncertainty could in principle reconcile our plateau AAE values higher than 1 with the common assumption that AAE tends to 1 when the aerosol is dominated by BC. Nevertheless, we modeled AAE as a function of aerosol chemical and microphysical properties (Fig.1). Such simulation clearly shows that AAE close to 1 can be obtained for aerosol population dominated by fresh fossil fuel combustion emissions, with diameter centered below 20 nm. Conversely the size distribution of aerosol population investigated in this study is centered around 80-300 nm (Fig.1). In addition source apportionment studies performed in the Po valley show that carbonaceous aerosol, both in urban and rural sites, is strongly affected by wood burning emissions (Gilardoni et al., 2011; Larsen et al., 2012; Gilardoni et al., 2016). It follows that, even considering the possible bias due to the experimental uncertainty, the AAE plateauing at values higher than 1 for large BC-to-OA ratios is still is in agreement with model results and source apportionment data.

Taken together, these findings prove that BrC in the observed ambient aerosol shows $AAE_{467-660}$ = 3.2$\pm$0.9 with $k_{(530)}$ = 0.017$\pm$0.001, and occurs in particles in the droplet mode size range, enriched in ammonium nitrate and poor in BC, with a strong dependance on OA-to-BC ratios, $SSA_{530}$ being 0.98$\pm$0.01. In particular, when the bulk aerosol is dominated by the BrC droplet mode particles, the AAE is greater than 2.3$\pm$1 (median $\pm$ uncertainty), and BC-to-OA ratios are lower than 0.05$\pm$0.03.

## 4.2 A case study

We present here a case study (Figure 4) to show the main microphysical and chemical features of the BrC aerosol observed.

On the case study day (i.e., February $1^{st}$, 2013) the relative humidity was high (97.5$\pm$0.4%, against a mean value for the Winter campaign of 82 $\pm$ 14 %, and a maximum of 98 %), temperature averaged 2.8 $\pm$ 0.0 ° C (campaign mean value=3.5 $\pm$ 2.8 ° C), and aerosol liquid water content was above 200 $\mu$g $\cdot$ m$^{-3}$ (Gilardoni et al., 2016). Absorption and scattering coefficients at 530 nm (Fig.1 of the supplementary material) ranged from 5 to 10 Mm$^{-1}$ (with larger AAE), and from 300 to 400 Mm$^{-1}$ (with lower Scattering Angstrom Exponents, SAE) respectively, with SSA$_{530}$=0.98$\pm$0.01. The number concentration of 2-10 $\mu$m particles (Fig.4e) had a peak at approx. 4:00 a.m. UTC (we interpret this as particle growth by water vapor), and then decreased until 9:00 a.m. UTC. After this, the number concentration of 0.3-1 $\mu$m particles increased. This increase occurred just after the part of the day (from 11:00 a.m. to 2:00 p.m) when the strong signal in the aerosol vertical profile at the ground (the darker red layer in Fig. 4d) is observed to dissipate (we interpret this as droplet evaporation). These processes are consistent with the formation of the droplet mode aerosol (John, 1990; Meng and Seinfield, 1994; Seinfeld and Pandis, 2006) explaining the increase of the droplet mode aerosol scores (PC3) observed in the afternoon. The case study was selected during this period (1.5 hour from 5:30 to 7:00 p.m), and relevant data are shown in Fig.2, Fig.3, Fig.5, and Fig.1 of the supplementary material (in this Figure, the case study is the first day of the Winter field campaign).

During the case study (i.e., from 5:30 to 7:00 p.m on February $1^{st}$, 2013) we observed peculiar data. The AAE was significantly higher (3.4–5.3) than the median value (2.0$\pm$0.5 during both field campaigns, and 2.1$\pm$0.6 in the Winter). The volume size distribution (Fig. 4a) was narrow and monomodal, centred on the droplet mode (d$_m$ from 450 to 700 nm). Relevant mass size distributions of the main constituents of nr-PM$_1$ (NO$_3^-$, OA, and NH$_4^+$) were centered around 700 nm of the vacuum aerodynamic diameter (d$_{va}$) corresponding to about 500 nm in mobility diameter (for spherical particles in the continuum regime with $\rho_p = 1.4 g \cdot cm^{-3}$ (Seinfeld and Pandis, 2006)). In addition, the OA mass below d$_{va}$ = 300 nm was significantly lower than that of the droplet mode, especially when compared to the average field results (Fig. 4c). It is important to note that the collection efficiency of the HR-ToF-AMS is 50% for 600 nm particles and decreases for larger sizes (Liu et al., 2007). This explains the difference between the size distributions in Fig. 4a and Fig. 4b at larger sizes.

At the light of source apportionment study performed on organic aerosol (Gilardoni et al., 2016) we explain the increase of AAE during the case study period with the formation of secondary organic aerosol in the aqueous phase associated with aerosol particles. The analysis of microphysical properties reported in this study confirms that the aqueous secondary organic aerosol formation adds mass to the atmospheric aerosol in the droplet mode range. This case study both illustrates and confirms general features observed for BrC during the whole field campaign.

## 4.3 Discussion in comparison with previous works

In this section we discuss our findings and explore their consistency with the literature.

The analysis of chemical and microphysical properties shows that BrC associated with the formation of secondary aerosol has a narrow monomodal size distribution centered around the droplet mode (400–700 nm) in the entire $PM_{10}$ size range. This result agrees with the observations reported by Lin et al. (2010), showing that 80% of the mass of atmospheric humic-like substances, a light absorbing organic aerosol component, was found in the droplet mode. The correspondence between
BrC and the droplet mode aerosol points to the important role that aqueous reactions within aerosol particles can play in the formation of light absorbing organic aerosol (Ervens et al., 2011; Laskin et al., 2015).

The study of optical and chemical properties indicates that in our case the larger AAE values associated with BrC depend on the organic fraction in a different way from that in literature (Shinozuka
et al., 2009; Russell et al., 2010; Arola et al., 2011). In Figure 5a we compare our measurements collected at the urban background site of Bologna with the trend previously found based on airborne and sunphotometers observations (Shinozuka et al., 2009; Russell et al., 2010). AAE is plotted versus the mass fraction of organic aerosol ($f_{OA}$), marker colour being $SSA_{530}$ as in Shinozuka et al. (2009)'s Fig.7. We show the best fit lines (thin black lines) identified by Shinozuka et al. (2009) and corre-
sponding to different SSA bins, and the best fit reported by Russell et al. (2010) (thick black line). Note that the larger AAE values in our study (associated to BrC in the droplet mode) correspond to Shinozuka et al. (2009)'s fit line at SSA=0.98–1, but were associated to lower scattering coefficients in those previous studies. There is consistency between our study and those reported previously. However, our data show that increasing AAE is accompanied by increasing the OA normalized to
BC (Fig.5b, r=0.78, p<0.001), rather than by increasing OA (Fig.5a). This comparison adds to the literature that for the ambient aerosol in the lower troposphere AAE correlates with OA-to-BC ratios far more than with the organic fraction.

We found that BrC corresponds to BC-to-OA ratios below 0.05±0.03. Fig.5c shows the dependence of AAE on the BC-to-OA ratio as parametrized by Saleh et al. (2014) and Lu et al. (2015)
for biomass burning emissions and primary organic aerosol emissions. The best fit line to measurements performed in this study (grey line) is similar to the fitting lines reported previously (thicker and thinner black lines, showing respectively Saleh et al. (2014)'s and Lu et al. (2015)'s data). While we cannot compare absolute values because we compare the AAE of the bulk aerosol (this study) to the AAE of OA only (Saleh et al., 2014; Lu et al., 2015), it is evident that patterns are similar.
This comparison extends the dependence of AAE on BC-to-OA observed in chamber experiments (solely for fresh biomass burning primary OA, and not for fossil fuel OA) by Saleh et al. (2014) and Lu et al. (2015) to ambient aerosol dominated by wood burning emissions (Gilardoni et al., 2016).

## 5   Summary and conclusions

We investigated optical-chemical-microphysical properties of brown carbon (BrC) in the urban ambient atmosphere. In situ ground measurements of chemical (HR-ToF-AMS), optical (3-$\lambda$ nephelometer and PSAP), and microphysical (SMPS and APS) aerosol properties were carried out in the Po Valley (Bologna), together with ancillary observations. BrC was identified and characterized by linking the wavelength dependence of light absorption (as indicated by the Absorption Ångström Exponent, AAE) in the visible region to key aerosol types with known size distributions, and to major PM$_1$ chemical components (BC, OA, nitrate, ammonium, and sulfate). BrC measurements were interpreted through numerical simulations (Mie theory) of AAE($d_p,\lambda,m$) resolved by particle size ($d_p$) and wavelength ($\lambda$) dependent complex refractive index (m$_{(\lambda)}$=n$_{(\lambda)}$-ik$_{(\lambda)}$) in the visible region.

We found that:

(1) AAE increases with increasing the (optically relevant) aerosol size. Larger AAEs (3.2$\pm$0.9, with values up to 5.5) occur when the bulk aerosol size distribution is dominated by the droplet mode, i.e. the large accumulation mode originating from the processing in the aqueous phase. These values identify BrC.

(2) Specific $m_{(\lambda)}$ values are necessary in addition to a proper particle size range to match the high AAE measured for BrC. These $m_{(\lambda)}$ values are theoretically expected to be: $k_{(467)} = 0.026\pm0.001$, $k_{(530)} = 0.017\pm0.001$, $k_{(660)} = 0.014\pm0.001$, and n$_{467}$=1.47$\pm$0.01 (SSA$_{530}$=0.98$\pm$0.01), consistent to literature $m_{(\lambda)}$ values for BrC in the ambient atmosphere.

(3) AAE increases with increasing the OA-to-BC ratio, rather than with increasing f$_{OA}$, the larger AAEs (and thus BrC) corresponding to larger ammonium nitrate (f$_{NO3}$=0.38$\pm$0.05, f$_{NH4}$=0.14$\pm$0.01) and lower BC (f$_{BC}$=0.01$\pm$0.01).

In the paper by Gilardoni et al. (2016) we investigated organic aerosol source and identified SOA formation from processing of biomass burning emissions in the aqueous phase. We than discussed the climate implication of this aqSOA formation at the light of its optical properties, including AAE. In the present paper we investigated the particle size distribution and spectral optical properties of brown carbon in the ambient aerosol, and related these to major aerosol chemical components. By combining the analysis of mycrophisical and optical properties reported here with the source apportionment study by Gilardoni et al. (2016), we demonstrated that the aqSOA formation adds mass to the atmospheric aerosol in the droplet mode range.

When exploring consistency of these findings with the literature, our study:

(i) provides experimental evidence that the size distribution of BrC associated with the formation of secondary aerosol is dominated by the droplet mode, consistent with recent findings pointing to the role of aqueous reactions within aerosol particles in the formation of BrC;

(ii) adds to sunphotometric observations (e.g., AERONET) that in the lower troposphere AAE correlates with the organic aerosol normalized to BC (i.e., OA-to-BC) far more than with the organic fraction;

(iii) extends to the ambient aerosol dominated by wood burning emissions the dependence of AAE on BC-to-OA previously observed in combustion chamber experiments.

These findings are expected to bear important implications for atmospheric modeling studies and remote sensing observations. Both BrC number size distribution and the dependence of AAE on BC-to-OA can be relevant to parametrize and investigate BrC in the ambient atmosphere. Findings can

be used to infer preliminary chemical information from optical information, as optical techniques are increasingly used to characterise aerosol properties.

*Acknowledgements.* This work was realized in the framework of the SUPERSITO Project, financed by the Emilia-Romagna Region (under the DR 1971/13). The work was partly accomplished in the framework of the DIAPASON ("Desert-dust Impact on Air quality through model-Predictions and Advanced Sensors Observa-

tioNs") project, funded by the European Commission (LIFE+ 2010 ENV/IT/391).

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

# TABLES

**Table 1.** Statistically significant (p<0.001) Pearson's correlation coefficients (r) between: Absorption Ångström Exponent (AAE) at 467-660 nm; scores of major aerosol types identified by PCA (PC1 is the road traffic related aerosol, PC2 is the residential heating related aerosol, and PC3 is the droplet mode aerosol); mass concentration and mass fractions ($f_x$) of Black Carbon (BC), organics (OA), nitrate ($NO_3^-$), sulfate ($SO_4^{2-}$), and ammonium ($NH_4^+$); optically relevant aerosol size representative of the entire aerosol population (calculated as median mobility diameter of the particle surface size distribution, $d_{med(S)}$); OA-to-BC ratios (see Tab.1 and Tab. 2 of the Supplementary material for additional relevant values).

| r | AAE | $d_{med(S)}$ | BC | $f_{BC}$ | OA | $f_{OA}$ | OA-to-BC | $NO_3$ | $f_{NO3}$ | $SO_4$ | $f_{SO4}$ | $NH_4$ | $f_{NH4}$ |
|---|---|---|---|---|---|---|---|---|---|---|---|---|---|
| $d_{med(S)}$ | 0.60 | 1 | -0.24 | -0.38 | – | -0.68 | 0.37 | 0.48 | 0.50 | 0.30 | 0.25 | 0.49 | 0.69 |
| AAE | 1 | 0.60 | -0.26 | -0.52 | 0.40 | -0.34 | 0.78 | 0.60 | 0.40 | 0.67 | 0.18 | 0.65 | 0.44 |
| PC1 | – | – | 0.56 | – | 0.83 | – | – | 0.52 | – | – | – | 0.52 | – |
| PC2 | – | -0.52 | – | – | – | 0.31 | – | -0.19 | – | – | – | -0.20 | -0.27 |
| PC3 | 0.66 | 0.60 | -0.38 | -0.53 | 0.12 | -0.52 | 0.54 | 0.46 | 0.50 | 0.45 | – | 0.49 | 0.58 |

**FIGURES**

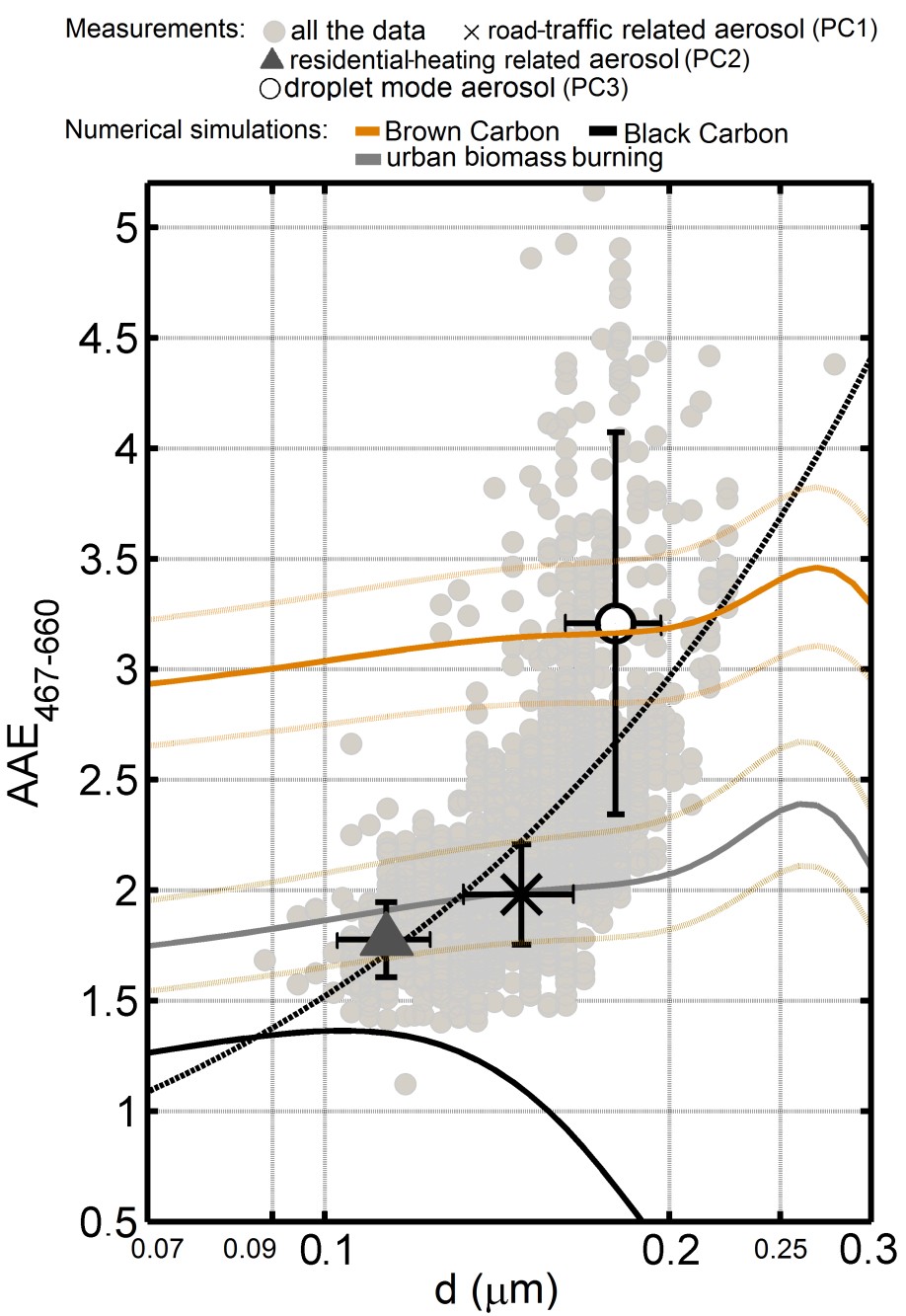

**Figure 1.** Experimentally measured and numerically simulated (Mie theory) relation between Absorption Ångström Exponent (AAE) at 467-660 nm and aerosol size (d). For measurements: (i) d is represented by the mobility median diameter of the $PM_{10}$ particle surface size distribution ($d_{med(S)}$); (ii) all the data measured are indicated by light grey markers, the dotted black line representing the best fit to these data; (iii) major aerosol types identified (Sect.3.3) are indicated by darker markers (median $\pm$ standard deviation). For numerical simulations (Sect.3.4): patterns theoretically expected for BrC, BC, and urban biomass burning are indicated by brown, black and grey thick lines respectively, the dotted thinner lines indicating the uncertainty of the refractive index ($m_{(\lambda)} = n_{(\lambda)}-ik_{(\lambda)}$) set to $\pm 0.01$ for $n_{(\lambda)}$ and $\pm 0.001$ for $k_{(\lambda)}$).

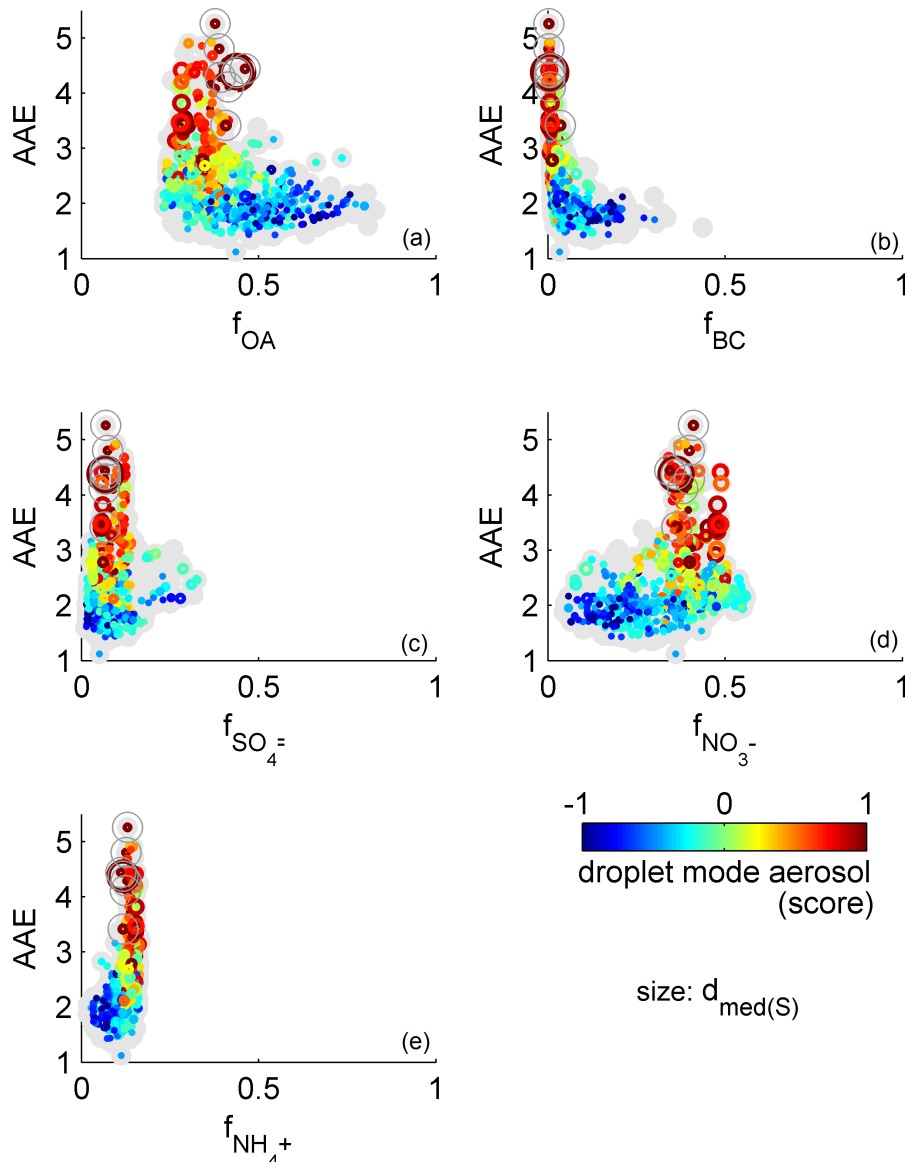

**Figure 2.** Physicochemical features of Brown Carbon. Absorption Ångström Exponent at 467-660 nm (AAE) plotted against mass fractions ($f_x$) of: (a) Organic Aerosol (OA), (b) Black Carbon (BC), (c) sulfate ($SO_4^{2-}$), (d) nitrate ($NO_3^-$), and (e) ammonium ($NH_4^+$). Grey markers show the longest available time series for AAE and $f_x$, while marker color and size indicate datapoints for which the droplet mode aerosol scores and $d_{med(S)}$ were available (Sect.3). Data indicated by dark grey "o" show case study values illustrated in Fig. 4.

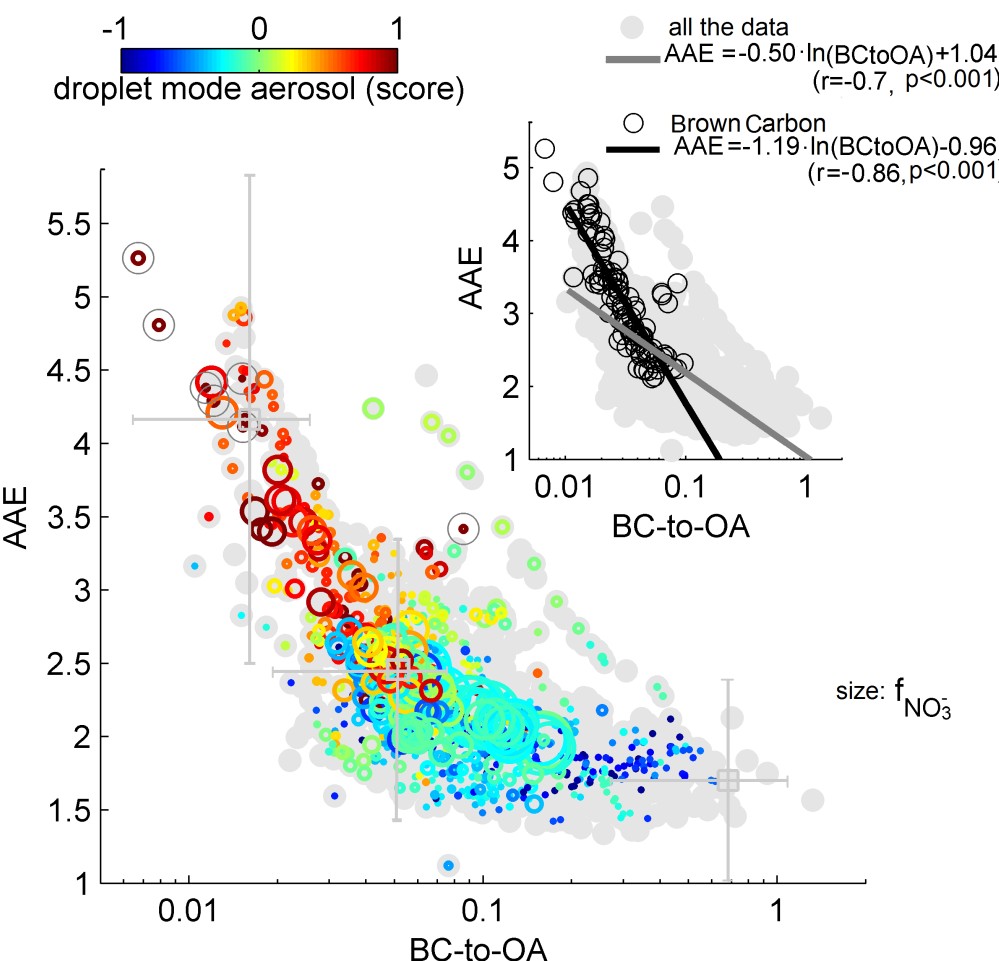

**Figure 3.** Dependence of AAE on BC-to-OA ratios. Grey markers show the longest available time series for AAE at 467-660 nm and BC-to-OA, while marker color and size indicate datapoints for which the droplet mode aerosol scores and $f_{NO3}$ were available (Sect.3). Median values (grey squares) and relevant data uncertainty are indicated at the upper, mean, and lower AAE bins. Data indicated by dark grey "o" show case-study values illustrated in Fig. 4. Inner panel: best fit lines with relevant Eq. and Pearson's correlation coefficients (r, p) to all the data measured (grey line) and droplet mode BrC data (black line) (as indicated by droplet mode aerosol score > 0).

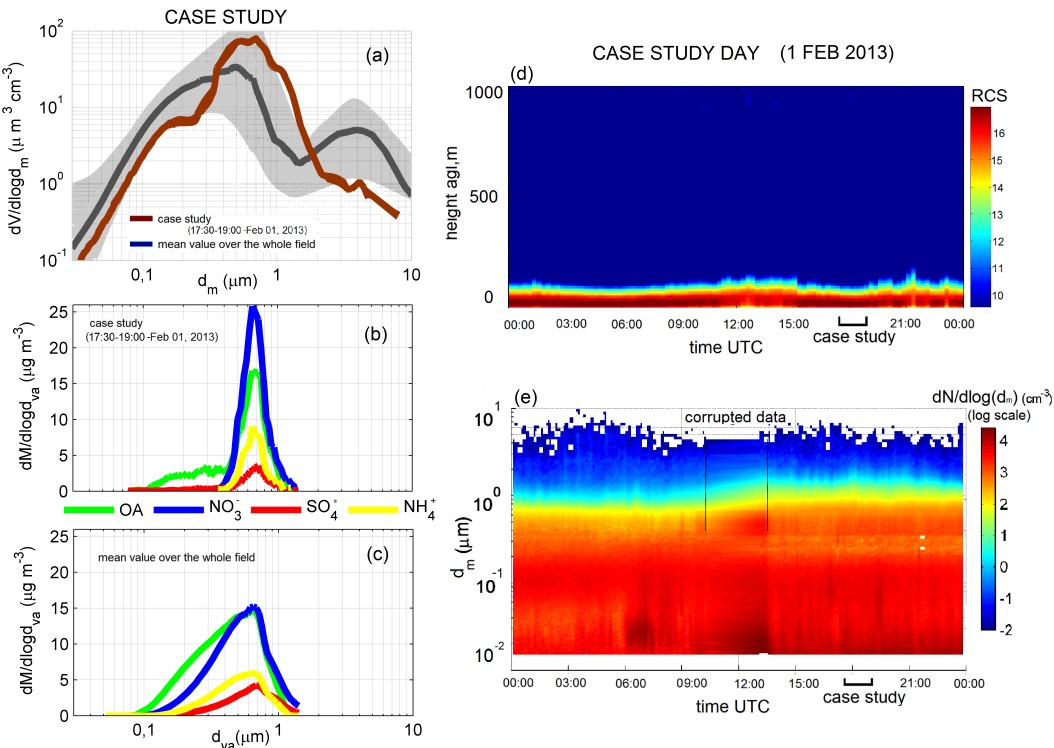

**Figure 4.** A case study illustrating BrC major features. Case study time period (1.5 hours) is from 17:30 to 19:00 UTC on February $1^{st}$, 2013 (Local Time at the Po Valley sampling site in this period is UTC + 1 hour). Case study values are compared with mean values over the whole field experiment. Panels illustrate: (a) particle volume size distribution (dV/dlog$_{10}$d$_m$, based on electrical mobility particle diameter d$_m$) during the case study and relevant mean values; (b) particle mass size distribution (dM/dlog$_{10}$d$_{va}$, based on vacuum aerodynamic diameter, d$_{va}$) during the case study, and (c) relevant mean values; (d) aerosol vertical profiles in the atmosphere during the entire case study day (time-height cross section of the range corrected signal, RCS=ln(S×$R^2$), from an LD40 ceilometer); (e) particle number size distributions during the entire case study day.

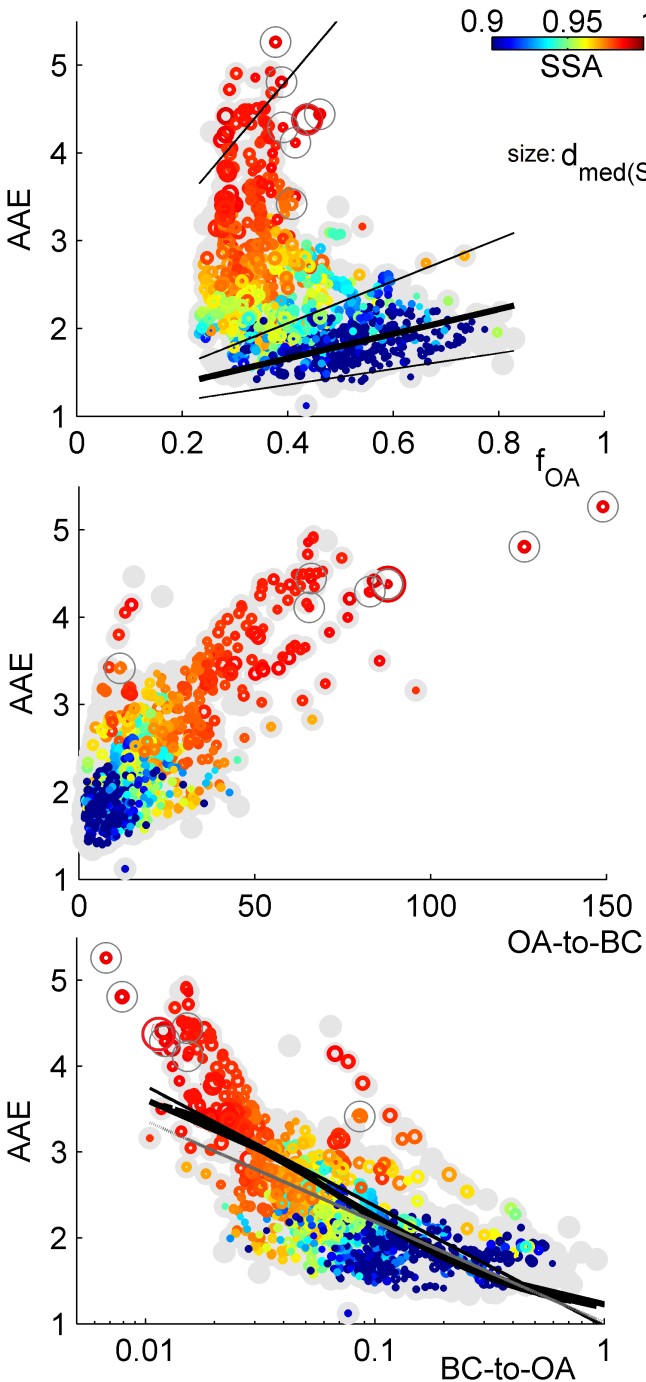

**Figure 5.** Absorption Ångström Exponent at 467-660 nm (AAE) against: (a) OA mass fraction, (b) OA-to-BC ratios, and (c) BC-to-OA ratios. Grey markers show the longest available time series for AAE and $f_{OA}$, while coloured markers show datapoints for which the SSA at 530nm and $d_{med_S}$ were available (Sect.3). Best fit lines to data taken from: (i) Shinozuka et al. (2009) (at different SSA bins, i.e. 0.98-1.00, 0.96-0.98, 0.90-0.92, from top to bottom), and Russell et al. (2010) are indicated in panel a (thin and thick black lines respectively); (ii) Saleh et al. (2014), and Lu et al. (2015) are indicated in panel c (black thick and thin lines, respectively); (iii) this work are indicated in panel c (grey line, as in Fig.3). Data indicated by dark grey "o" show case-study values illustrated in Fig. 4.