# Peer review of "Characteristics of Brown Carbon in the urban Po Valley atmosphere"

_Atmospheric Chemistry and Physics, 2015_

## Referee Comment (RC1) · Anonymous Referee #1 · 23 Feb 2016

This manuscript by Costabile et al. reports field measurements of aerosol microphysical, chemical, and optical properties in the Po Valley. Combining the measurements with statistical analysis, the authors conclude that brown carbon (light-absorbing organic aerosol (OA)) is mostly associated with aged OA. They also show that the spectral-dependence of the aerosol light-absorption – the Absorption Ångström Exponent (AAE) – increases with the increasing mass ratio of OA to black carbon (OA-to-BC ratio), and suggest that this confirms previous literature findings.

The measurements and the analysis are interesting and timely, however, I have major concerns (see major comments below) regarding the validity of part of the analysis and the associated conclusions. I believe that this paper can potentially be suitable for publication in ACP if the authors adequately address these issues.

**Major comments:**

1. BrC is mainly composed of aged OA

This is one of the main conclusions in this paper. The authors arrive at this conclusion based on the correlations they found between AAE and OA-to-BC ratio, and between AAE and f44. The premise that this conclusion is based on is that AAE is an indicator of how brown the aerosol is. This is not accurate. AAE is an indicator of the wavelength-dependence of absorption. What is more "brown," OA with MAC=2 $m^2$/g (at 532nm) and AAE=2, or OA with MAC=0.2$m^2$/g (at 532nm) and AAE=4? Probably the brownest OA reported in the literature is that observed by Alexander et al. (*Science 321*, 833-836 [2008]), and it had relatively small AAE.

So, with their current analysis, the authors can only make conclusions on the wavelength-dependence of OA absorption, and not on how brown the OA is. However, even this needs further analysis to be convincing. The authors base their analysis on the AAE of the whole aerosol (including both BC and OA). The AAE of the aerosol is dictated by the relative contribution of the components (and coating effects, which we can set aside for now). The increase of AAE with increasing OA-to-BC ratio does not necessarily mean that the added OA upon aging has a larger AAE, it can simply mean that the relative contribution of BC to AAE gets smaller, so the overall AAE increases. In other words, fresh and aged OA could have similar AAE, and the increase in AAE vs OA-to-BC ratio is simply due to the decreased contribution of BC.

2. Validity of PSAP measurements and BC concentration calculations

Figure 4 and figure 6 show that AAE is ~2 at BC-to-OA ratio of 20. This puts in question the validity of AAE retrieved from the measurements because at such large BC-to-OA ratio, the

particles should have AAE closer to 1. The authors should address this issue to determine whether there is something wrong with the calculations or the measurements that resulted in this factor of 2 overestimation of AAE. Also, BC concentrations are largely underestimated because the calculations (section 3.1) rely on the assumption that AAE of BC is equal to 1 (which is not the case, as retrieved from the instrument in this study). Consequently, the fractions of the different components should be recalculated.

**Specific comments:**

P1 L15: What's the difference between brown OA and brown carbon?

P2 L1: What do you mean by "moderately volatile?" Also, how do you support this claim?

P5 L12: "we fixed threshold…" please explain.

P6 L24: It is not clear why this assumption is needed or whether it is valid. You need more than "mass scales with volume" to justify this assumption.

P7 top paragraph: What are PC2 and PC4? What would be in the primary aerosol that has modes 20-100 nm and 10-40 nm? One would expect those to be BC particles.

Section 5.1: The authors mention that they will compare results with Shinozuka et al. (2009) Russell et al. (2010), Arola et al. (2011), Saleh et al. (2014), and Lu et al. (2015) but only show comparison with Russell et al. and Saleh et al.

Section 5.1: The AAE values in this study are much larger than Russell et al., especially at low $f_{OA}$ values. I believe this is due to a bias in AAE retrieval (see major comment 2).

Section 5.1: Comparison with Saleh et al. is not apples-to-apples. Saleh et al. reports the wavelength-dependence of the imaginary part of the refractive index of the OA only, while this study reports AAE (wavelength-dependence of the absorption coefficient) of the total aerosol (including BC).

Section 5.1: The finding that AAE in this study is twice that in Saleh et al. and Lu et al. is most probably not accurate due to overestimation of AAE in this study (see major comment 2).

Terminology: the authors go back and forth between BrC, brown, and "brown." Please be consistent.

**Technical corrections:**

There are many grammatical/sentence structure errors (I just list a few below). The manuscript should be edited for language before it can publishable.

Title: delete "an".

P1

L2: Angstrom should be Ångström (here and elsewhere).

L4: Delete "a" before "brown".

L5: "Composed by" should be "composed of".

L5: Change "enriched in" to "contain".

---

## Referee Comment (RC2) · Anonymous Referee #2 · 29 Feb 2016

**General Comments**

This paper presents the results from ambient aerosol measurements carried out in the Po Valley. Chemical and microphysical properties of the aerosol were measured – the focus of this study is on the observation and characterization of brown carbon (BrC), which the authors attribute to SOA. This conclusion is mostly based upon a strong correlation between the AAE and the OA/BC ratio. Overall, this study is novel and the findings will contribute to the growing body of work characterizing atmospheric BrC. There are some major issues that will have to be addressed before the manuscript is suitable for ACP, but I have confidence that the authors can address these comments (detailed below).

**Specific Comments**

- The concept of measurement uncertainty has not been addressed at all in the manuscript. This needs to be added throughout – especially for the measurements that are central to the analyses: AAE and OA/BC ratio (or BC-to-OA ratio, as in Fig. 4).

- Similarly, the QA/QC procedures and standards need to be discussed and defended. E.g., all data with $\sigma_a < 1$ Mm$^{-1}$ and $\sigma_s < 10$ Mm$^{-1}$ were removed – why were these the limits imposed? How much data were discarded? In Section 3, what is meant by "The dataset consisted of 11211 records (5764 in fall, and 5447 in winter), including 2551 records (covering 40 days of measurements) with no missing value, and 1087 records (150 in fall, and 937 in winter) of cleaned data after data analysis."? Does this mean only ~10% of the measurements were ultimately included in this analysis? Section 3.2 – what constituted an acceptable merge of the SMPS and APS size distributions? What was considered unacceptable? How many measurements were eliminated using the smoothing procedure and visual inspection?

- Section 4.1 and Figures 2-4: discussion is needed to explain the physical meaning of the PC score and the factor loading numbers.

- Section 4.1 – what does the following sentence mean? "BC mass concentration was assumed to increase mostly with increasing concentration of larger BC particles…"

- Section 4.1 – what does the following sentence mean? "Higher fBC values coupled to lower BC mass concentration were, therefore, interpreted as indicators of ultrafine BC particles, and vice versa."

- Figure 1: it is not clear why the present results from the Po Valley are compared to results from Leipzig made 7-8 years ago?

- The authors have missed some other relevant work that also shows associations between ambient SOA and BrC – see for example X. Zhang et al. (2011; 2013).

- Section 5.2 and Figure 7: although the 'paradigm' discussed here may have been developed in a prior paper, it is not something I think most readers will be familiar with (this reviewer was not). Provide the necessary explanation and context to interpret the present results.

- Section 5.2, lines 28-34: perhaps this is in line with the above comment, but I was completely confused by this entire passage.

- In my opinion, Figure 8 does not much at all to the paper – I would recommend removing it.

- Section 6: the findings do not "prove" the formation of BrC in the atmosphere.

- Pg. 7, line 21-22: "The dependence on the nitrate mass fraction ($f_{NO3}$ , Fig.3d) is not obvious, as high AAE values and droplet mode scores are observed for both $f_{NO3}$ <0.05 and $f_{NO3}$ '0.25." This does not seem consistent with the discussion of nitrate's importance in the abstract or in Section 6.

**Technical Corrections**

- Pg. 3, line 10: change to "Following the approach of Saleh et al. (2014),…"

**References**

Zhang, X., Lin, Y.-H., Surratt, J. D., Zotter, P., Prevot, A. S. H., Weber, R. J., Light‐absorbing soluble organic aerosol in Los Angeles and Atlanta: A contrast in secondary organic aerosol, *Geophys. Res. Lett.*, 38, L21810, 2011.

Zhang, X., Lin, Y.-H., Surratt, J. D., Weber, R. J., Sources, Composition and Absorption Angstrom Exponent of Light-absorbing Organic Components in Aerosol Extracts from the Los Angeles Basin, *Environ. Sci. Technol.*, 47(8), 2013.

---

## Author Comment (AC1) · 6 May 2016

We sincerely thank the referee for giving valuable and critical comments. These comments, all considered and addressed, have improved the work and strengthened its findings. Please, see detailed author's responses below.

(In Bold, comments from the referee; plain text is used for author's answers; in Italic, the new text added in the revised manuscript, including page and line numbers (Px,Lx-x).)

**1. BrC is mainly composed of aged OA**

**This is one of the main conclusions in this paper. The authors arrive at this conclusion based on the correlations they found between AAE and OA-to-BC ratio, and between AAE and f44. The premise that this conclusion is based on**

**is that AAE is an indicator of how brown the aerosol is. This is not accurate. AAE is an indicator of the wavelength-dependence of absorption. What is more "brown" OA with MAC=2 m2/g (at 532nm) and AAE=2, or OA with MAC=0.2m2/g (at 532nm) and AAE=4? Probably the brownest OA reported in the literature is that observed by Alexander et al. (Science 321, 833-836 [2008]), and it had relatively small AAE. So, with their current analysis, the authors can only make conclusions on the wavelength-dependence of OA absorption, and not on how brown the OA is. However, even this needs further analysis to be convincing. The authors base their analysis on the AAE of the whole aerosol (including both BC and OA). The AAE of the aerosol is dictated by the relative contribution of the components (and coating effects, which we can set aside for now). The increase of AAE with increasing OA-to-BC ratio does not necessarily mean that the added OA upon aging has a larger AAE, it can simply mean that the relative contribution of BC to AAE gets smaller, so the overall AAE increases. In other words, fresh and aged OA could have similar AAE, and the increase in AAE vs OA-to-BC ratio is simply due to the decreased contribution of BC.**

We agree that the conclusion **BrC is mainly composed of aged OA** is not **accurate**. In fact, we conversely concluded (and this is stressed in the revised manuscript) that: (i) *we observed the bulk nondust aerosol having the strongest spectral dependance of light absorption (AAE=2.5–6)*; (ii) AAEs of this aerosol indicate that it has brown color, but *not necessarily that it equates to brown carbon*; (iii) this aerosol with brown color is defined "brown" aerosol; (iv) *this "brown" aerosol has a secondary origin, large concentrations of organic aerosol (OA), high OA-to-BC ratios, and is dominated by accumulation mode particles*. We revised the manuscript accordingly.

In details:

(1) We modified the title: *Characteristics of brown aerosol in the urban Po Valley atmosphere*

[Figure]

(2) In Sect. 4.1, we stressed that the secondary origin of the "brown" aerosol is inferred from *a strong correlation found with the droplet mode aerosol type, a submode of accumulation mode aerosol particles which is by definition secondary in origin* (formed through secondary processes in the atmosphere). This conclusion is reinforced by the correlation with $f_{44}$.

(3) In Sect. 4.1, we stressed that the correlation with the droplet mode aerosol *gives insights into the likely formation process of this "brown" aerosol*.

(4) In Sect. 5.3, we stressed that there are indications that *this "brown" aerosol contains aged OA*. Thus, we speculate *into aged brown OA formation processes*.

(5) In Sect. 5.1, we stressed that we did not directly measure the AAE of OA, but *the AAE of the bulk dust-free aerosol, including boh BC and OA contributions* ($AAE_{BC,OA}$). Changes in this AAE are due to changes in *relative proportions of BC and OA*. This was already illustrated in former Figure 3b of the manuscript, where AAE did not linearly decrease with decreasing $f_{BC}$. We added Supplementary Figure 1 showing relative contributions of OA (data color) and BC (data size) on AAE.

(6) In Sect. 5.1, we stressed that our findings indicate *patterns (not absolute values) of $AAE_{OA}$*, found to *increase with decreasing BC-to-OA, similarly to what observed by Saleh et al. (2014) and Lu et al. (2015)*. To this purpose, we added Supplementary figure 2 showing AAE measured (i.e., $AAE_{BC,OA}$) together with AAE used to calculate BC ($AAE_{BC}$) in the AAE vs BC-to-OA chart. (For this figure, please, consider the following answers to referee's comments **Comparison with Saleh et al. is not apples-to-apples...**, and **Validity of PSAP measurements and BC concentration calculations**).

**2. Validity of PSAP measurements and BC concentration calculations.**

**Figure 4 and figure 6 show that AAE is 2 at BC-to-OA ratio of 20. This puts in question the validity of AAE retrieved from the measurements because at such**

**large BC-to-OA ratio, the particles should have AAE closer to 1. The authors should address this issue to determine whether there is something wrong with the calculations or the measurements that resulted in this factor of 2 overestimation of AAE. Also, BC concentrations are largely underestimated because the calculations (section 3.1) rely on the assumption that AAE of BC is equal to 1 (which is not the case, as retrieved from the instrument in this study). Consequently, the fractions of the different components should be recalculated.**

We agree that this important issue was not illustrated enough, and revised the manuscript accordingly to include: (1) PSAP measurement uncertainty (Sect.2.2), and (2) BC mass concentration calculation method and resulting uncertainty (Sect.3.1); (3) these uncertainties in relevant figures; and (4) the explanation that (Sect.4.1) the low AAE values the referee refers to, can be due to these uncertainties and relevant (low) OA values.

In details:

(1) BC was calculated from PSAP-derived absorption coefficients ($\sigma_a$), relevant uncertainties *deriving from a number of measurement artifacts.* We referenced previous works in literature which have developed correction algorithms to overcome these artifacts, and mentioned that, based on Virkkula et al. (2010)'s correction, $\sigma_a$ uncertainty is $\frac{\delta(\sigma_a)}{\sigma_a}$ = 0.2.

(2) *BC mass concentration was calculated using the AAE$_{BC}$ attribution method.* This method *assumes a known value of AAE$_{BC}$ at 530-660 nm.* This value is *commonly set to 1 for externally mixed BC.* We referenced previous works in literature, in particular Lack and Langridge (2013) and references therein, which have showed this AAE$_{BC}$ to be (i) larger than 1 for BC aerosol in ambient air (including both internally and externally mixed BC); (ii) *for externally mixed BC, 1 for particles with diameter < 50 mm, and 0.8–1.1 for diameters of 50-200 nm; (iii) 0.55–1.7 for internally mixed BC, depending on particle size, coating, core, wavelengths.* Based on these previous works, we

used $AAE_{BC}=1.1$, and $\frac{\delta(AAE_{BC})}{AAE_{BC}}=0.22$. (Note that there is no **fraction of the different components** that **should be recalculated**, as no fraction due to Brown Carbon was derived.)

(3) Figure 4 was revised. Both this figure and Suppl. Fig.1, include BC and AAE uncertainties, calculated propagating these $\sigma_a$ and $AAE_{BC}$ uncertainties.

(4) We explained *the small trend of the lower AAE values toward 1.5* considering: (i) uncertainties before mentioned, and (ii) OA values ($>0$) these low AAE values are at. On one hand, Fig.1 shows that AAE values from 1 to 2 are within measurement uncertanties. On the other hand, OA values $> 0$ in Suppl. Fig.3 suggest that there may have been *spectrally light absorbing material that the AMS cannot detect (refractory material, or material in particles smaller than 100 nm and larger than 1 $\mu m$)* causing a bias in AAE toward values larger than 1. We referenced previous papers in literature discussing this topic. In particular, Lack et al. (2008) analyzed the "bias in filter-based aerosol light absorption measurements due to organic aerosol loading", and found that "at low OA concentrations" (similar to OA concentrations here), "where HOA was a larger fraction of the OA, PSAP-derived AAE did display a small upward trend to $\sim 1.5$, suggesting that the HOA measured may be mildly absorbing".

**P1 L15: What's the difference between brown OA and brown carbon?** The referee is right for questioning this. The lack of a common definition for brown carbon causes confusion. We do believe that the scientific community should address this question.

**P2 L1: What do you mean by "moderately volatile"? Also, how do you support this claim?** We referred to the classification proposed by Pöschl (2003) - Figure 1 of their paper. In the revised manuscript, we referenced this work based on which *there is a gradual decrease of thermochemical refractiveness and specific optical absorption going from BC/EC graphite-like structures to non-refractive and colorless OC. Also, there is a gradual decrease in the volatility from BC (the lowest volatility), to colorless non-refractive volatile OC (the highest volatility). A broad range of coloured organic*

*compounds, with volatility in between these two extremes, have recently emerged .* (These were referred to as "moderately volatile".)

**P5 L12: "we fixed threshold"? please explain.** We explained in the revised manuscript (Par.3.1) that *dust-free aerosol conditions were identified based on the analysis of aerosol spectral optical properties.* We referenced previous works in literature, which have used this analysis to gather information on aerosol type. Among these, Costabile et al. (2013) showed that *the aerosol dominated by dust can be unambiguously identified based on the following combination of SAE, AAE, SSA, and dSSA: $SSA_{530} > 0.85, SAE_{467-660} < 0.5 , AAE_{467-660} \sim 2, dSSA_{660-467}=0.05-0.3.$* We used this method to identify cases when the aerosol was dominated by dust, and eliminate them.

**P6 L24: It is not clear why this assumption is needed or whether it is valid. You need more than "mass scales with volume" to justify this assumption** We agree that this assumption is not needed, and deleted lines at P6L23-28 to avoid unnecessary text.

**P7 top paragraph: What are PC2 and PC4? What would be in the primary aerosol that has modes 20-100 nm and 10-40 nm? One would expect those to be BC particles.**

The referee is right for noting that PC2 and PC4 are both BC primary aerosols. However, their source is different (probably, heating and traffic emissions, respectively). We revised relevant text.

In details:

(1) We added Supplementary Figure 3, to illustrate weekly diurnal cycles, and loadings of PC1-4.

(2) We clarified (P7L3-5) that *PC1, PC2, and PC4 are BC primary aerosols (all correlated to BC and $f_{BC}$). PC1 and PC4 are both sourced by traffic emissions, diurnal*

[Figure]

*cycles peaking at rush hours and week-days. PC1 is a fine mode aerosol component (PNSD mode peaking from 100 to 200 nm), PC4 an ultrafine aerosol component (PNSD peaking from 20 to 40 nm). PC2 is an ultrafine BC aerosol component, as PC4. Unlike PC4, however, PC2 higher scores are at night-time, and there is no weekly cycle and no peak at "rush hours": PC2 is probably sourced by heating emissions.*

**Section 5.1: The authors mention that they will compare results with Shinozuka et al. (2009) Russell et al. (2010), Arola et al. (2011), Saleh et al. (2014), and Lu et al. (2015) but only show comparison with Russell et al. and Saleh et al.**

These comparisons (both reinforcing our findings) were missing, indeed. We revised Sect.5.1 accordingly.

In details:

(1) Shinozuka et al. (2009)'s results were added in panel a of the revised Figure 6. Relevant data for SSA bins of 0.90-0.92, 0.96-0.98 and 0.98-1.00 were indicated by grey lines.

(2) Lu et al. (2015)'s results were added by grey line in panel c of the revised Figure 6. It was indicated that y-axis of both Lu et al. (2015)'s and Saleh et al (2014)'s results refers to $AAE_{OA}$, while y-axis of our figure refers to AAE from both BC and OA contributions.

(3) We added that *values of AAE>3.5 are at SSA > 0.98, consistent with Shinozuka et al. (2009)'s results at SSA= 0.98-1.00.* We added that *this comparison ultimately shows consistency of "brown" aerosol properties measured in situ at the ground and observed from airborne and AERONET.*

(4) Although we are consistent with Arola et al. (2011)'s findings, the comparison in Fig.6 with their work is not possible. They used absorbing OC column concentrations [mg m$^{-2}$] in the x-axis, whereas we show $f_{OA}$ derived from in situ ground measurements. We therefore did not reference this work.

**Section 5.1: The AAE values in this study are much larger than Russell et al., especially at low fOA values. I believe this is due to a bias in AAE retrieval (see major comment 2).** We agree that there are differences, but there is broad consistency, as well. This is more clear in the revised Figure 6, where Shinozuka et al. (2009)'s grey lines (based on the same Russell et al. (2010)'s data) reveal the strong dependence on SSA. This was less clearly indicated by Russell et al. (2010)'s black line.

**Section 5.1: Comparison with Saleh et al. is not apples-to-apples. Saleh et al. reports the wavelength-dependence of the imaginary part of the refractive index of the OA only, while this study reports AAE (wavelength-dependence of the absorption coefficient) of the total aerosol (including BC).** The referee is correct for mentioning this issue, which we addressed by previous comments as well. We revised the manuscript accordingly.

In details:

(1) Figure 6 was revised to show that Saleh et al. (2014)'s and Lu et al.(2015)'s curves represent $AAE_{OA}$.

(2) Sect.5.1 was revised to point out that *we compare, in Fig.6c, different AAE values. In our work, AAE values are due to the bulk dust-free aerosol, thus depending on both BC and OA ($AAE_{BC,OA}$). AAE values of Saleh et al. (2014) and Lu et al. (2015) ($AAE_{OA}$) come from the wavelength-dependence of OA alone ($w_{OA}$), excluding contributions from BC ($AAE_{OA}$ values are calculated from $w_{OA}$ based on the relation AAE=w+1, valid for small particles in the visible range). The comparison in Fig.6c is thus intended to compare patterns only, not absolute values.*

(3) The Supplementary Figure 2 was added to show $AAE_{BC}$ together with $AAE_{BC,OA}$, in the AAE vs BC-to-OA plot. We mentioned that *this figure is intended to indicate wavelenght dependence of OA absorption, and to suggest possible $AAE_{OA}$ patterns, increasing with decreasing BC-to-OA, similarly to what observed by Saleh et al. (2014)*

*and Lu et al. (2015).*

**Section 5.1: The finding that AAE in this study is twice that in Saleh et al. and Lu et al. is most probably not accurate due to overestimation of AAE in this study (see major comment 2).** We agree, and revised Sect.5.1 of the manuscript to mention that *the AAE of this secondary "brown" aerosol would probably be higher than that of the fresh biomass burning OA it likely derives from* (not "twice that", but "higher than that").

**Terminology: the authors go back and forth between BrC, brown, and "brown". Please be consistent.** As previously mentioned, *"brown" aerosol does not necessarily equate to brown carbon.* In the revised manuscript, we stressed this and used the *"brown" aerosol* notation everywhere to indicate this aerosol type. BrC was substituted by brown carbon.

**Technical corrections** have all been addressed.

**References**

Lack, D.A., Cappa, C.D., Covert, D.S., Baynard, T., Massoli, P., Sierau, B., Bates, T.S., Quinn, P.K., Lovejoy, E.R. and Ravishankara, A.R.: Bias in filter-based aerosol light absorption measurements due to organic aerosol loading: Evidence from ambient measurements. Aerosol Science and Technology, 42(12), pp.1033-1041, 2008.

Lack, D. A. and Langridge, J. M.: On the attribution of black and brown carbon light absorption using the Ångström exponent, Atmos. Chem. Phys., 13, 10535-10543, doi:10.5194/acp-13-10535-2013, 2013.

Pöschl, U.: Aerosol particle analysis: challenges and progress. Anal. Bioanal. chem., 375.1, 30-32, DOI 10.1007/s00216-002-1611-5, 2003.

**Fig. 1.** Revised Figure 4. *Relation between "brown" aerosol and Black Carbon (BC) to Organic Aerosol (OA) ratio (BC-to-OA). Absorption Angstrom Exponent at 467-660 nm (AAE) is plotted against BC-to-OA. Data color is the score of the droplet mode PC extracted by the statistical analysis. Data size is the median diameter of the particle surface size distribution ($d_{med(S)}$). Data indicated by "*" show case-study values illustrated in Fig. 5. Grey bars indicate measurement uncertainty.*

**Fig. 2.** Revised Figure 6. *Dependence of Absorption Angstrom Exponent at 467-660 nm (AAE) on (a) organic aerosol mass fraction ($f_{OA}$), (b) Organic Aerosol (OA) to Black Carbon (BC) ratio (OA-to-BC), and (c) BC-to-OA ratio. Data color is Single Scattering Albedo at 530 nm. Data size is median diameter of particle surface size distributions ($d_{med_S}$, ranging from 50 to 300 nm). Data indicated by "*" show case study values illustrated in Fig. 5. Relevant Pearson's correlation coefficients (r) are indicated. For comparison, we show previous results: (a) by Russell et al. (2010) (black line) and Shinozuka et al. (2009) (grey lines, for the SSA bins of 0.90-0.92, 0.96-0.98 and 0.98-1.00) ; (c) by Saleh et al. (2014) (black line), and Lu et al. (2015) (grey line). Note that AAE includes contributions from both BC and OA, whereas in panel (c), Lu et al. (2015) Saleh et al. (2014)'s results refer to $AAE_{OA}$ only.*

**Fig. 3.** Supplementary Figure 1. *Relative proportions of BC and OA in the AAE vs BC-to-OA relation. BC-to-OA ratio (x-axis) against Absorption Angstrom Exponent (y-axis). Data points are sized by BC mass fraction ($f_{BC}$), and colored by organic aerosol mass concentration (OA). Vertical and horizontal bars show uncertainties*

**Fig. 4.** Supplementary Figure 2. *Dependance of the Absorption Angstrom Exponent (AAE) due to organic aerosol (OA) from the BC-to-OA ratio. The AAE due to BC and OA ($AAE_{BC,OA}$) of the dust-free bulk AAE is indicated by black squares. (A subset of data is indicated with uncertainty-bars (grey lines), the interpolating curve of the whole dataset being indicated by the black curve). $AAE_{BC}$ used to derive BC is also indicated (grey squares).*

**Fig. 5.** Supplementary Figure 3. *The four principal components (PC1-PC4) of the Particle Number Size Distribution: (a-d) weekly diurnal cycles of scores (red, Week Day; green, Week End; black, all data), and (e) loadings.*

[Figure]

**Fig. 6.** Revised Figure 4.

[Figure]

**Fig. 7.** Revised Figure 6.

[Figure]

**Fig. 8.** Supplementary Figure 1.

Fig. 9. Supplementary Figure 2.

[Figure]

Fig. 10. Supplementary Figure 3.

---

## Author Comment (AC2) · 6 May 2016

We would like to thank the referee for his constructive and precious comments. These comments have all been considered and addressed, and have significantly improved upon the work. Please, see our detailed author's responses below.

(In Bold, comments from the Referee; plain text is used for author's answers ; in Italic, the new text added in the revised manuscript, including page and line numbers (Px,Lx-x).)

**The concept of measurement uncertainty has not been addressed at all in the manuscript. This needs to be added throughout - especially for the measurements that are central to the analyses: AAE and OA/BC ratio (or BC-to-OA ratio, as in Fig. 4). Similarly, the QA/QC procedures and standards need to be dis-**

[Figure]

**cussed and defended. E.g., all data with $\sigma_a <$ 1 Mm$^{-1}$ and $\sigma_s <$ 10 Mm$-1$ were removed : why were these the limits imposed? How much data were discarded?**

We revised the manuscript to address: (1) uncertainties of optical variables in Sect.2.2; (2) uncertainty of OA in Sect.2.3; (3) uncertainty of BC in Sect.3.1; (4) the resulting uncertainties in the AAE vs BC-to-OA relation in Sect.4.1 (Figure 4 was revised); (5) QA/QC procedures and standards in Sect.2. We referenced previous papers in literature these uncertainties were based on.

In details:

(1) *The uncertainty of the scattering coefficient $\frac{\delta(\sigma_s)}{\sigma_s}$ = 0.02, and that of the absorption coefficient $\frac{\delta(\sigma_a)}{\sigma_a}$ = 0.2.*

(2) *The uncertainty of the AMS-derived OA $\frac{\delta(OA)}{OA}$ = 0.2.*

(3) The uncertainty of BC was calculated by propagating $\frac{\delta(\sigma_a)}{\sigma_a}$, together with the uncertainty deriving from the AAE$_{BC}$ attribution method used to calculate BC. In this method, we set AAE$_{BC}$ =1.1, and $\frac{\delta(AAE_{BC})}{AAE_{BC}}$=0.22.

(4) Uncertainties of AAE and BC-to-OA were calculated propagating these values. Figure 4 of the manuscript was revised to include these uncertainties.

(5) We mentioned that *outlier/low values can significantly influence data statistics*. We hence checked carefully values of: (i) $\sigma_a <$ 1 Mm$^{-1}$ and $> 50$ Mm$^{-1}$, as *PSAP sensitivity $<$1 Mm$^{-1}$ in the measurement range 0-50 Mm$^{-1}$; (ii) $\sigma_s <$ 10 Mm$^{-1}$ and $> 1000$ Mm$^{-1}$, as nephelometer lower detectable limit= 0.3 Mm$^{-1}$, with calibration tolerance of $\pm$ 4 Mm$^{-1}$, in the measurement range 0-2000 Mm$^{-1}$.* These values are set by instrument manufacturers. *A few data (124 records having $\sigma_a <$1 Mm$^{-1}$, less than 20 records with $\sigma_s <$10 Mm$^{-1}$, and some points with $\sigma_s >$700 Mm$^{-1}$) were discarded, as they were considered dubious values, comparing to data variability during the field* (illustrated in Supplementary Figure 1).

**In Section 3, what is meant by "The dataset consisted of 11211 records (5764 in fall, and 5447 in winter), including 2551 records (covering 40 days of measurements) with no missing value, and 1087 records (150 in fall, and 937 in winter) of cleaned data after data analysis."? Does this mean only 10% of the measurements were ultimately included in this analysis?**

The referee is right for noting that there was an error. Text was corrected as it follows:

*Dataset consisted of 3211 records (1764 in fall, and 1447 in winter). The statistical analysis was done on a subset of these data with no empty field (2551 records, covering 40 days of measurements). These data were then cleaned, and a final dataset of 1487 records (550 in fall, and 937 in winter) was ultimately included in the statistical analysis. The longer dataset was, however, used in the analysis to evaluate single cases (e.g., the case-study).*

**Section 3.2 : what constituted an acceptable merge of the SMPS and APS size distributions? What was considered unacceptable? How many measurements were eliminated using the smoothing procedure and visual inspection?**

The referee correctly mentions an issue that was addressed in the revised version (Sect.3.2, P5L24-25).

In details:

*$PNSD_{fitted}$ replaced $PNSD_{APS}$ and $PNSD_{SMPS}$ when their relative difference ($d_r$, Eq.1):*

$$d_r = \frac{|PNSD_{SMPS} - PNSD_{APS}|}{max[PNSD_{SMPS}, PNSD_{APS}]} \qquad (1)$$

*was larger than 0.1 cm$^{-3}$. This procedure was considered acceptable if: (i) the minimum mean squared error between $PNSD_{fitted}$ and $PNSD_{APS}$ was less than 1%; (ii) correlation coefficients between $PNSD_{fitted}$ and $PNSD_{SMPS}$, and between $PNSD_{fitted}$*

*and $PNSD_{APS}$ were larger than 0.8. A number of 98 records did not verify these conditions, and were checked by visual inspection: 94 of them were discarded, and 4 accepted.*

**Section 4.1 and Figures 2-4: discussion is needed to explain the physical meaning of the PC score and the factor loading numbers.** We agree with the referee, and added this in the revised manuscript.

In details:

*Coefficients of $PC_k$ represent the relative weight (in terms of correlation) of original variables (i.e.,time series of dN/dlog($d_p$)) in $PC_k$. Factor loadings of $PC_k$ represent these coefficients scaled by the variance explained by $PC_k$ ($\varkappa_k$) . Loadings of $PC_k$ thus represent the relative weight of the dN/dlog($d_p$) variables in $PC_k$ re-scaled by $\varkappa_k$ . Factor scores of $PC_k$ are the transformed variables corresponding to a particular data point in the dN/dlog($d_p$) time series. Factor scores thus represent $PC_k$ values corresponding to each particular data point of the dN/dlog($d_p$) time series.*

**Section 4.1 : what does the following sentence mean? "BC mass concentration was assumed to increase mostly with increasing concentration of larger BC particles"? Section 4.1 ? what does the following sentence mean? "Higher fBC values coupled to lower BC mass concentration were, therefore, interpreted as indicators of ultrafine BC particles, and vice versa."**

Both referees (and editor, as well) indicated that this sentence is both unclear and not necessary. To avoid unnecessary text, we decided to delete it (P6L23-28).

**Figure 1: it is not clear why the present results from the Po Valley are compared to results from Leipzig made 7-8 years ago?** We revised the manuscript (Sect.4.1) to clarify that: (1) this comparison aims at reinforcing *the interpretation that the aerosol type represented by PC3 is the droplet mode aerosol; and (2) the correlation found between the droplet mode aerosol and the "brown" aerosol demostrates that the "brown"*

*aerosol is secondary in origin (the droplet mode is secondary in origin), and gives insights into its likely formation process.*

In details:

(1) To our knowledge, results from Leipzig are *the only work in literature showing a similar aerosol principal component to compare with*. The statistical methodology of the two works is the same, but datasets are very different, in time and space. Results from Leipzig were *based on five different statistical analysis based on different datasets* (two year data at eight concurrent measurement sites were available to create these five different dataset to be analysed). *Results were correlated to meteorological and air quality data.* Our work here was based on a shorter time period, and one measurement site. We do believe that results from Leipzig are statistically strong. These identified clearly the *PC representing the droplet mode aerosol*, which shows *broad similarities with PC3* obtained in our work. *This allows to deduce with a reasonable statistical accuracy that PC3 does represent the droplet mode aerosol.*

**The authors have missed some other relevant work that also shows associations between ambient SOA and BrC - see for example X. Zhang et al. (2011; 2013).**

The referee is right for mentioning that these are important works, both supporting our findings. In fact, we referenced the work by Zhang et al. (2013) in the former manuscript (P2L12, P10L13-15). In the revised manuscript, we referenced these works more explicitly.

In details:

(1) In Sect.1 (P2L12-13), we indicated that *secondary organic aerosol (SOA) formed in the atmosphere contributes to the light absorbing carbon, as well (Moise et al., 2015), but only a few works have analysed secondary brown carbon associated to SOA (Zhang et al., 2013, 2011), and Saleh et al.(2013).*

(2) In Sect.5.3 (P10L13-15), we mentioned that *the composition found for this "brown"*

*aerosol (high OA, nitrates being a likely component), its formation process (involving aqueous phase reactions), and AAEs values, are all coherent with previous studies, which showed increased light absorption towards UV for SOA particles (Jacobson, 1999; Lee et al., 2013; Song et al., 2013; Zhang et al., 2013; Powelson et al., 2014; Lin et al., 2014; Laskin et al., 2015), and sources, composition, and AAE of light-absorbing soluble organic aerosol in urban areas (Zhang et al., 2013, 2011)* .

**Section 5.2 and Figure 7: although the "paradigm" discussed here may have been developed in a prior paper, it is not something I think most readers will be familiar with (this reviewer was not). Provide the necessary explanation and context to interpret the present results.** We agree with the comment. In the revised manuscript, we added in Sect.5.2: (1) the context (including references to previous relevant works which have analysed the topic); (2) additional explanations to interpret results (including the revised Figure 7), and (3) Table 2 summarising these results (in particular, $k_{530}$, which is compared to relevant values of bulk aged OA in literature).

In details:

(1) We indicated that *cluster analysis of aerosol spectral optical properties is becoming more and more used to infer information on aerosol type from optical data.* We mentioned that *Costabile et al. (2013) assessed spectral optical properties of key aerosol populations through Mie theory: soot, biomass burning, two types of organics, dust and marine particles were simulated through a sectional approach where each of these aerosol types was given a monomodal PNSD and a set of three refractive indices (RIs) in the visible range. Relevant Angstrom Exponents of extinction, scattering, and absorption (EAE, SAE, AAE), SSA and its spectral variation (dSSA) were calculated. It was proved that these aerosol types separately cluster within a "paradigm" where SAE is on the y-axis, dSSA times AAE is on the x-axis, and SSA is on the z-axis.*

(2) We indicated that *experimental data of the "brown" aerosol do cluster in this paradigm* (Fig. 7), and that *the cluster of "brown" aerosol data is separated from all*

*other simulated aerosol types, except that named "large organics". Data of "large organics" and "brown" aerosol do overlap, indicating that they may represent the same aerosol type. In fact, microphysical properties of the aerosol type named "large organics" were simulated to be same as those of the droplet mode aerosol (i.e., PNSD peaking in the "large" accumulation mode, 300-800 nm size range). Spectral optical properties of this "large organics" aerosol type were simulated by RIs of spectrally absorbing organic material in the visible region: these RIs, given the broad similarities in Fig.7, can be assumed to be those of the "brown" aerosol.*

(3) We added Table 2 summarizing these results, and pointed that *this comparison ultimately allows to infer spectral optical properties of "brown" aerosol* (in particular, $k_{530}$, compared to relevant values recently reviewed by Lu et al. (2015) ).

**- Section 5.2, lines 28-34: perhaps this is in line with the above comment, but I was completely confused by this entire passage.** This text has been changed (see previous comment).

**- In my opinion, Figure 8 does not much at all to the paper ? I would recommend removing it.** We agree with the referee that this figure is not needed. However, it may be useful to connect results from different aerosol communities. As well, it reinforce findings. We would therefore prefer to keep it.

**- Section 6: the findings do not "prove" the formation of BrC in the atmosphere.** We agree with the referee, and revised the text (P6L22-23) to mention that *findings show that there is "brown" aerosol in the atmosphere*.

**- Pg. 7, line 21-22: "The dependence on the nitrate mass fraction (fNO3 , Fig.3d) is not obvious, as high AAE values and droplet mode scores are observed for both fNO3 <0.05 and fNO3 '0.25." This does not seem consistent with the discussion of nitrate's importance in the abstract or in Section 6.** The referee is right and we modified text accordingly, indicating (P1L4-5), that *findings show that "brown" aerosol... contains large concentrations of organic aerosol (OA) in droplet mode parti-*

*cles* and that *Nitrate is an additional likely component.*

Technical corrections have been addessed.

**References**

Zhang, X., Y.-H. Lin, J. D. Surratt, and R. J. Weber: Sources, composition and absorption angström exponent of light-absorbing organic components in aerosol extracts from the Los Angeles Basin, Environ. Sci. Technol., 47(8), 3685-3693, 2013.

Zhang, X., Lin, Y. H., Surratt, J. D., Zotter, P., Prévˆt, A. S., Weber, R. J.:. Light?absorbing soluble organic aerosol in Los Angeles and Atlanta: A contrast in secondary organic aerosol. Geophys. Res. Lett., 38(21), 2011.

———————————————————

[Figure]

**Fig. 1.** Revised Figure 4. *Relation between "brown" aerosol and Black Carbon (BC) to Organic Aerosol (OA) ratio (BC-to-OA). Absorption Angstrom Exponent at 467-660 nm (AAE) is plotted against BC-to-OA. Data color is the score of the droplet mode PC extracted by the statistical analysis. Data size is the median diameter of the particle surface size distribution ($d_{med(S)}$). Data indicated by "\*" show case-study values illustrated in Fig.5. Grey bars indicate measurement uncertainty, added to a subset of data only*.

**Fig. 2.** Revised Figure 7. *Optical signature of the "brown aerosol" as indicated by the paradigm proposed by Costabile et al. (2013). Absorption Angstrom Exponent at 467-660 nm (AAE) times spectral variation of Single Scattering Albedo from 660 to 467 nm (dSSA = $SSA_{660}$ - $SSA_{467}$) is plotted against Scattering Angstrom Exponent at 467-660 nm (SAE). Experimental data of the droplet mode obtained in this work (representing the "brown" aerosol) are compared with key aerosol types obtained through Mie simulations. Data color is SSA at 520 nm. Relevant AAE values for key aerosol types are indicated as mean±standard deviation*.

**Fig. 3.** Supplementary Figure. *Spectral optical properties during the two measurement fields: y-axis shows data of scattering and absorption coefficients, colored by the Scattering Angstrom Exponent (SAE), and Absorption Angstrom Exponent (AAE), respectively.*

[Figure]

**Fig. 4.** Revised Figure 4.

☆ experimental data (this work)
○ simulated data (Costabile et al., 2013)

SSA

"pure"soot
AAE=1.2±0.01
k$_{530}$=0.047

AAE=3.0±0.02
small organics
k$_{530}$=0.011

biomass
burning
AAE=2.1±0.01
k$_{530}$=0.021

large inorganics
AAE=2.5±0.03
k$_{530}$=0.00001
k$_{467}$=0.00002

large organics
AAE=2.4±0.03
k$_{530}$=0.008
k$_{467}$=0.012

"brown" aerosol
AAE=3.5±0.8

marine
AAE=3.1±0.1
k$_{530}$=0.00006

k$_{530}$=0.004
dust
AAE=1.67±0.04

SAE

dSSA× AAE

**Fig. 5.** Revised Figure 7.

**Fig. 6.** Supplementary Figure 1.

**Table 1.** Spectral optical properties of the "brown" aerosol in the visible range: AAE, SSA, and SAE (expressed as mean±standard deviation and variation range [minimum- maximum value]), and real and imaginary part ($k_\lambda$) of the complex refractive index (RI [$\lambda$]).

| $AAE_{467-660}$ | $SSA_{530}$ | $SAE_{467-660}$ | $RI_\lambda$ real part | $k_\lambda$ | $[\lambda]$ |
|---|---|---|---|---|---|
| | | | 1.460 | $1.2 \cdot 10^{-2}$ | [467nm] |
| 3.5±0.8 [2.5-6] | 0.97±0.01 [0.92-0.99] | 0.8±0.3 [0-2] | 1.454 | $8 \cdot 10^{-3}$ | [530nm] |
| | | | 1.512 | $7.5 \cdot 10^{-3}$ | [660nm] |

**Fig. 7.** Table 2.

---

## Author Response (AR1)

Dear editor,

we would like to thank you for having supported a very constructive review of our manuscript. We do believe that the work has improved significantly.

Please, find below our point-by-point responses to the reviews (as published on 6 May 2016). These include: (I) comments from Referees (in Bold), (II) author's response (plain text), (III) author's changes in manuscript (in Italic).

All relevant changes made in the manuscript are showed in the marked-up manuscript version attached at the end of this document.

**1 Referee 1**

(I) Comment from Referee:

**1. BrC is mainly composed of aged OA**

**This is one of the main conclusions in this paper. The authors arrive at this conclusion based on the correlations they found between AAE and OA-to-BC ratio, and between AAE and f44. The premise that this conclusion is based on is that AAE is an indicator of how brown the aerosol is. This is not accurate. AAE is an indicator of the wavelength-dependence of absorption. What is more "brown" OA with MAC=2 m2/g (at 532nm) and AAE=2, or OA with MAC=0.2m2/g (at 532nm) and AAE=4? Probably the brownest OA reported in the literature is that observed by Alexander et al. (Science 321, 833-836 [2008]), and it had relatively small AAE. So, with their current analysis, the authors can only make conclusions on the wavelength-dependence of OA absorption, and not on how brown the OA is. However, even this needs further analysis to be convincing. The authors base their analysis on the AAE of the whole aerosol (including both BC and OA). The AAE of the aerosol is dictated by the relative contribution of the components (and coating effects, which we can set aside for now). The increase of AAE with increasing OA-to-BC ratio does not necessarily mean that the added OA upon aging has a larger AAE, it can simply mean that the relative contribution of BC to AAE gets smaller, so the overall AAE increases. In other words, fresh and aged OA could have similar AAE, and the increase in AAE vs OA-to-BC ratio is simply due to the decreased contribution of BC.**

(II) Author's response.

We agree that the conclusion **BrC is mainly composed of aged OA** is not **accurate**. In fact, we conversely concluded (and this is stressed in the revised manuscript) that: (i) *we observed the bulk nondust aerosol having the strongest spectral dependance of light absorption (AAE=2.5–6)*; (ii) AAEs of this aerosol indicate that it has brown color, but *not necessarily that it equates to brown carbon*; (iii) this aerosol with brown color is defined "brown" aerosol; (iv) *this "brown" aerosol has a secondary origin, large concentrations of organic aerosol (OA), high OA-to-BC ratios, and is dominated by accumulation mode particles*. We revised the manuscript accordingly.

(III) Changes in manuscript.

(*) We modified the title: *Characteristics of brown aerosol in the urban Po Valley atmosphere*

(*) In Sect. 4.1, we stressed that the secondary origin of the "brown" aerosol is inferred from *a strong correlation found with the droplet mode aerosol type, a submode of accumulation mode aerosol particles which is by definition secondary in origin* (formed through secondary processes in the atmosphere). This conclusion is reinforced by the correlation with $f_{44}$.

(*) In Sect. 4.1, we stressed that the correlation with the droplet mode aerosol *gives insights into the likely formation process of this "brown" aerosol.*

(*) In Sect. 5.3, we stressed that there are indications that *this "brown" aerosol contains aged OA.* Thus, we speculate *into aged brown OA formation processes.*

(*) In Sect. 5.1, we stressed that we did not directly measure the AAE of OA, but *the AAE of the bulk dust-free aerosol, including boh BC and OA contributions* ($AAE_{BC,OA}$). Changes in this AAE are due to changes in *relative proportions of BC and OA*. This was already illustrated in former Figure 3b of the manuscript, where AAE did not linearly decrease with decreasing $f_{BC}$. We added

Supplementary Figure 1 showing relative contributions of OA (data color) and BC (data size) on AAE.

(*) In Sect. 5.1, we stressed that our findings indicate *patterns (not absolute values) of AAE$_{OA}$*, found to *increase with decreasing BC-to-OA, similarly to what observed by Saleh et al. (2014) and Lu et al. (2015)*. To this purpose, we added Supplementary figure 2 showing AAE measured (i.e., AAE$_{BC,OA}$) together with AAE used to calculate BC (AAE$_{BC}$) in the AAE vs BC-to-OA chart.

(*) Our findings indicate *patterns (not absolute values) of AAE$_{OA}$*, found to *increase with decreasing BC-to-OA, similarly to what observed by* ? *and* ?. To this purpose, we added the following Supplementary figure (Fig. 6 here) showing AAE measured (i.e., AAE$_{BC,OA}$) together with AAE used to calculate BC (AAE$_{BC}$) in the AAE vs BC-to-OA chart.

(I) Comment from Referee:

**2. Validity of PSAP measurements and BC concentration calculations.**

**Figure 4 and figure 6 show that AAE is 2 at BC-to-OA ratio of 20. This puts in question the validity of AAE retrieved from the measurements because at such large BC-to-OA ratio, the particles should have AAE closer to 1. The authors should address this issue to determine whether there is something wrong with the calculations or the measurements that resulted in this factor of 2 overestimation of AAE. Also, BC concentrations are largely underestimated because the calculations (section 3.1) rely on the assumption that AAE of BC is equal to 1 (which is not the case, as retrieved from the instrument in this study). Consequently, the fractions of the different components should be recalculated.**

(II) Author's response.

We agree that this important issue was not illustrated enough, and revised the manuscript accordingly to include: (i) PSAP measurement uncertainty (Sect.2.2), and (ii) BC mass concentration calculation method and resulting uncertainty (Sect.3.1); (iii) these uncertainties in relevant figures; and (iv) the explanation that (Sect.4.1) the low AAE values the referee refers to, can be due to these uncertainties and relevant (low) OA values.

(III) Changes in manuscript.

(i) BC was calculated from PSAP-derived absorption coefficients ($\sigma_a$), relevant uncertainties *deriving from a number of measurement artifacts*. We referenced previous works in literature which have developed correction algorithms to overcome these artifacts, and mentioned that, based on Virkkula et al. (2010)'s correction, $\sigma_a$ uncertainty is $\frac{\delta(\sigma_a)}{\sigma_a} = 0.2$.

(ii) *BC mass concentration was calculated using the AAE$_{BC}$ attribution method*. This method *assumes a known value of AAE$_{BC}$ at 530-660 nm*. This value is *commonly set to 1 for externally mixed BC*. We referenced previous works in literature, in particular Lack and Langridge (2013) and references therein, which have showed this AAE$_{BC}$ to be (i) larger than 1 for BC aerosol in ambient air (including both internally and externally mixed BC); (ii) *for externally mixed BC, 1 for particles with diameter < 50 mm, and 0.8–1.1 for diameters of 50-200 nm; (iii) 0.55–1.7 for internally mixed BC, depending on particle size, coating, core, wavelengths*. Based on these previous works, we used $AAE_{BC}=1.1$, and $\frac{\delta(AAE_{BC})}{AAE_{BC}}=0.22$.

(iii) Figure 4 was revised. Both this figure and Suppl. Fig.1, include BC and AAE uncertainties, calculated propagating these $\sigma_a$ and $AAE_{BC}$ uncertainties.

(iv) We explained *the small trend of the lower AAE values toward 1.5* considering: (i) uncertainties before mentioned, and (ii) OA values (>0) these low AAE values are at. On one hand, Fig.1 shows that AAE values from 1 to 2 are within measurement uncertanties. On the other hand, OA values > 0 in Suppl. Fig.5 suggest that there may have been *spectrally light absorbing material that the AMS cannot detect (refractory material, or material in particles smaller than 100 nm and larger than 1 μm)* causing a bias in AAE toward values larger than 1. We referenced previous papers in literature discussing this topic. In particular, Lack et al. (2008) analyzed the "bias in filter-based aerosol light absorption measurements due to organic aerosol loading", and found that "at low OA concentrations" (similar to OA concentrations here), "where HOA was a larger fraction of the OA,

PSAP-derived AAE did display a small upward trend to $\sim 1.5$, suggesting that the HOA measured may be mildly absorbing".

(I) Comment from Referee:

**P1 L15: What's the difference between brown OA and brown carbon?**

(II) Author's response.

The referee is right for questioning this. The lack of a common definition for brown carbon causes confusion. We do believe that the scientific community should address this question.

(I) Comment from Referee:

**P2 L1: What do you mean by "moderately volatile"? Also, how do you support this claim?**

(II) Author's response.

We referred to the classification proposed by Pöschl (2003) - Figure 1 of their paper. In the revised manuscript, we referenced this work.

(III) Changes in manuscript.

We explained in the Introduction section that *there is a gradual decrease of thermochemical refractiveness and specific optical absorption going from BC/EC graphite-like structures to non-refractive and colorless OC. Also, there is a gradual decrease in the volatility from BC (the lowest volatility), to colorless non-refractory volatile OC (the highest volatility). A broad range of coloured organic compounds, with volatility in between these two extremes, have recently emerged .*

(I) Comment from Referee:

**P5 L12: "we fixed threshold"? please explain.**

(II) Author's response.

We explained the method in the revised manuscript. And finally said that we used this method to identify cases when the aerosol was dominated by dust, and eliminate them.

(III) Changes in manuscript.

In Par.3.1 we explained that: *dust-free aerosol conditions were identified based on the analysis of aerosol spectral optical properties.* We referenced previous works in literature, which have used this analysis to gather information on aerosol type. Among these, Costabile et al. (2013) showed that *the aerosol dominated by dust can be unambiguously identified based on the following combination of SAE, AAE, SSA, and dSSA: $SSA_{530} >0.85$, $SAE_{467-660} <0.5$ , $AAE_{467-660} \sim 2$, $dSSA_{660-467}=0.05$-$0.3$.*

(I) Comment from Referee:

**P6 L24: It is not clear why this assumption is needed or whether it is valid. You need more than "mass scales with volume" to justify this assumption**

(II) Author's response.

We agree that this assumption is not needed, and:

(III) Changes in manuscript.

we deleted lines at P6L23-28 to avoid unnecessary text.

(I) Comment from Referee:

**P7 top paragraph: What are PC2 and PC4? What would be in the primary aerosol that has modes 20-100 nm and 10-40 nm? One would expect those to be BC particles.**

(II) Author's response.

The referee is right for noting that PC2 and PC4 are both BC primary aerosols. However, their source is different (probably, heating and traffic emissions, respectively). We revised relevant text.

(III) Changes in manuscript.

(*) We added Supplementary Figure 3, to illustrate weekly diurnal cycles, and loadings of PC1-4.

(*) We clarified (P7L3-5) that *PC1, PC2, and PC4 are BC primary aerosols (all correlated to BC and $f_{BC}$). PC1 and PC4 are both sourced by traffic emissions, diurnal cycles peaking at rush*

*hours and week-days. PC1 is a fine mode aerosol component (PNSD mode peaking from 100 to 200 nm), PC4 an ultrafine aerosol component (PNSD peaking from 20 to 40 nm). PC2 is an ultrafine BC aerosol component, as PC4. Unlike PC4, however, PC2 higher scores are at night-time, and there is no weekly cycle and no peak at "rush hours": PC2 is probably sourced by heating emissions.*

(I) Comment from Referee:

**Section 5.1: The authors mention that they will compare results with Shinozuka et al. (2009) Russell et al. (2010), Arola et al. (2011), Saleh et al. (2014), and Lu et al. (2015) but only show comparison with Russell et al. and Saleh et al.**

(II) Author's response.

These comparisons (both reinforcing our findings) were missing, indeed. We revised Sect.5.1 accordingly.

(III) Changes in manuscript.

(*) Shinozuka et al. (2009)'s results were added in panel a of the revised Figure 6. Relevant data for SSA bins of 0.90-0.92, 0.96-0.98 and 0.98-1.00 were indicated by grey lines.

(*) Lu et al. (2015)'s results were added by grey line in panel c of the revised Figure 6. It was indicated that y-axis of both Lu et al. (2015)'s and Saleh et al (2014)'s results refers to $AAE_{OA}$, while y-axis of our figure refers to AAE from both BC and OA contributions.

(*) We added that *values of AAE>3.5 are at SSA > 0.98, consistent with Shinozuka et al. (2009)'s results at SSA= 0.98-1.00.* We added that *this comparison ultimately shows consistency of "brown" aerosol properties measured in situ at the ground and observed from airborne and AERONET.*

(*) Although we are consistent with Arola et al. (2011)'s findings, the comparison in Fig.6 with their work is not possible. They used absorbing OC column concentrations [mg m$^{-2}$] in the x-axis, whereas we show $f_{OA}$ derived from in situ ground measurements. We therefore did not reference this work.

(I) Comment from Referee:

**Section 5.1: The AAE values in this study are much larger than Russell et al., especially at low fOA values. I believe this is due to a bias in AAE retrieval (see major comment 2).**

(II) Author's response.

We agree that there are differences, but there is broad consistency, as well. This is more clear in the revised Figure 6, where Shinozuka et al. (2009)'s grey lines (based on the same Russell et al. (2010)'s data) reveal the strong dependence on SSA. This was less clearly indicated by Russell et al. (2010)'s black line.

(I) Comment from Referee:

**Section 5.1: Comparison with Saleh et al. is not apples-to-apples. Saleh et al. reports the wavelength-dependence of the imaginary part of the refractive index of the OA only, while this study reports AAE (wavelength-dependence of the absorption coefficient) of the total aerosol (including BC).**

(II) Author's response.

The referee is correct for mentioning this issue, which we addressed by previous comments as well. We revised the manuscript accordingly.

(III) Changes in manuscript.

(*) Figure 6 was revised to show that Saleh et al. (2014)'s and Lu et al.(2015)'s curves represent $AAE_{OA}$.

(*) Sect.5.1 was revised to point out that *we compare, in Fig.6c, different AAE values. In our work, AAE values are due to the bulk dust-free aerosol, thus depending on both BC and OA ($AAE_{BC,OA}$). AAE values of Saleh et al. (2014) and Lu et al. (2015) ($AAE_{OA}$) come from the wavelength-dependence of OA alone ($w_{OA}$), excluding contributions from BC ($AAE_{OA}$ values are calculated from $w_{OA}$ based on the relation AAE=w+1, valid for small particles in the visible range). The comparison in Fig.6c is thus intended to compare patterns only, not absolute values.*

(*) The Supplementary Figure 2 was added to show $AAE_{BC}$ together with $AAE_{BC,OA}$, in the AAE vs BC-to-OA plot. We mentioned that *this figure is intended to indicate wavelenght dependence of OA absorption, and to suggest possible $AAE_{OA}$ patterns, increasing with decreasing BC-to-OA, similarly to what observed by Saleh et al. (2014) and Lu et al. (2015).*

(I) Comment from Referee:

**Section 5.1: The finding that AAE in this study is twice that in Saleh et al. and Lu et al. is most probably not accurate due to overestimation of AAE in this study (see major comment 2).**

(II) Author's response.

We agree, and revised Sect.5.1 of the manuscript to mention that *the AAE of this secondary "brown" aerosol would probably be higher than that of the fresh biomass burning OA it likely derives from* (not "twice that", but "higher than that").

(I) Comment from Referee:

**Terminology: the authors go back and forth between BrC, brown, and "brown". Please be consistent.**

(II) Author's response.

As previously mentioned, *"brown" aerosol does not necessarily equate to brown carbon.* In the revised manuscript, we stressed this and used the *"brown" aerosol* notation everywhere to indicate this aerosol type. BrC was substituted by brown carbon.

**2 Referee 2**

**The concept of measurement uncertainty has not been addressed at all in the manuscript. This needs to be added throughout - especially for the measurements that are central to the analyses: AAE and OA/BC ratio (or BC-to-OA ratio, as in Fig. 4). Similarly, the QA/QC procedures and standards need to be discussed and defended. E.g., all data with $\sigma_a < 1$ Mm$^{-1}$ and $\sigma_s < 10$ Mm$-1$ were removed : why were these the limits imposed? How much data were discarded?**

(II)(III) Author's response and Changes in manuscript.

We revised the manuscript to address: (i) uncertainties of optical variables in Sect.2.2; (ii) uncertainty of OA in Sect.2.3; (iii) uncertainty of BC in Sect.3.1; (iv) the resulting uncertainties in the AAE vs BC-to-OA relation in Sect.4.1 (Figure 4 was revised); (v) QA/QC procedures and standards in Sect.2. We referenced previous papers in literature these uncertainties were based on. This is detailed below:

(i) *The uncertainty of the scattering coefficient $\frac{\delta(\sigma_s)}{\sigma_s} = 0.02$, and that of the absorption coefficient $\frac{\delta(\sigma_a)}{\sigma_a} = 0.2$.*

(ii) *The uncertainty of the AMS-derived OA $\frac{\delta(OA)}{OA} = 0.2$.*

(iii) The uncertainty of BC was calculated by propagating $\frac{\delta(\sigma_a)}{\sigma_a}$, together with the uncertainty deriving from the AAE$_{BC}$ attribution method used to calculate BC. In this method, we set AAE$_{BC}$ =1.1, and $\frac{\delta(AAE_{BC})}{AAE_{BC}}$=0.22.

(iv) Uncertainties of AAE and BC-to-OA were calculated propagating these values. Figure 4 of the manuscript was revised to include these uncertainties.

(v) We mentioned that *outlier/low values can significantly influence data statistics*. We hence checked carefully values of: (i) $\sigma_a < 1$ Mm$^{-1}$ and $> 50$ Mm$^{-1}$, as *PSAP sensitivity $<1$ Mm$^{-1}$ in the measurement range 0-50 Mm$^{-1}$; (ii) $\sigma_s < 10$ Mm$^{-1}$ and $> 1000$ Mm$^{-1}$, as nephelometer lower detectable limit= 0.3 Mm$^{-1}$, with calibration tolerance of $\pm$ 4 Mm$^{-1}$, in the measurement range 0-2000 Mm$^{-1}$.* These values are set by instrument manufacturers. *A few data (124 records having $\sigma_a <1$ Mm$^{-1}$, less than 20 records with $\sigma_s <10$ Mm$^{-1}$, and some points with $\sigma_s >700$ Mm$^{-1}$) were discarded, as they were considered dubious values, comparing to data variability during the field* (illustrated in Supplementary Figure 1).

**In Section 3, what is meant by "The dataset consisted of 11211 records (5764 in fall, and 5447 in winter), including 2551 records (covering 40 days of measurements) with no missing value, and 1087 records (150 in fall, and 937 in winter) of cleaned data after data analysis."? Does this mean only 10% of the measurements were ultimately included in this analysis?**

(II)(III) Author's response and Changes in manuscript.

The referee is right for noting that there was an error. Text was corrected as it follows:

*Dataset consisted of 3211 records (1764 in fall, and 1447 in winter). The statistical analysis was done on a subset of these data with no empty field (2551 records, covering 40 days of measurements). These data were then cleaned, and a final dataset of 1487 records (550 in fall, and 937 in winter) was ultimately included in the statistical analysis. The longer dataset was, however, used in the analysis to evaluate single cases (e.g., the case-study).*

 **Section 3.2 : what constituted an acceptable merge of the SMPS and APS size distributions? What was considered unacceptable? How many measurements were eliminated using the smoothing procedure and visual inspection?**

(II)(III) Author's response and Changes in manuscript.

The referee correctly mentions an issue that was addressed in the revised version (Sect.3.2, P5L24-25).

$PNSD_{fitted}$ replaced $PNSD_{APS}$ and $PNSD_{SMPS}$ when their relative difference ($d_r$, Eq.1):

$$d_r = \frac{|PNSD_{SMPS} - PNSD_{APS}|}{max[PNSD_{SMPS}, PNSD_{APS}]} \tag{1}$$

was larger than $0.1\ cm^{-3}$. This procedure was considered acceptable if: (i) the minimum mean squared error between $PNSD_{fitted}$ and $PNSD_{APS}$ was less than 1%; (ii) correlation coefficients between $PNSD_{fitted}$ and $PNSD_{SMPS}$, and between $PNSD_{fitted}$ and $PNSD_{APS}$ were larger than 0.8. A number of 98 records did not verify these conditions, and were checked by visual inspection: 94 of them were discarded, and 4 accepted.

(I) Comment from Referee:
**Section 4.1 and Figures 2-4: discussion is needed to explain the physical meaning of the PC score and the factor loading numbers.**

(II) Author's response.
We agree with the referee, and added this in the revised manuscript.

(III) Changes in manuscript.
The following text was added to sect.4.1:

Coefficients of $PC_k$ represent the relative weight (in terms of correlation) of original variables (i.e., time series of $dN/dlog(d_p)$) in $PC_k$. Factor loadings of $PC_k$ represent these coefficients scaled by the variance explained by $PC_k$ ($\varkappa_k$). Loadings of $PC_k$ thus represent the relative weight of the $dN/dlog(d_p)$ variables in $PC_k$ re-scaled by $\varkappa_k$. Factor scores of $PC_k$ are the transformed variables corresponding to a particular data point in the $dN/dlog(d_p)$ time series. Factor scores thus represent $PC_k$ values corresponding to each particular data point of the $dN/dlog(d_p)$ time series.

(I) Comment from Referee: **Section 4.1 : what does the following sentence mean? "BC mass concentration was assumed to increase mostly with increasing concentration of larger BC particles"? Section 4.1 ? what does the following sentence mean? "Higher fBC values coupled to lower BC mass concentration were, therefore, interpreted as indicators of ultrafine BC particles, and vice versa."**

(II)(III) Author's response and Changes in manuscript.
Both referees (and editor, as well) indicated that this sentence is both unclear and not necessary. To avoid unnecessary text, we decided to delete it (P6L23-28).

(I) Comment from Referee:
**Figure 1: it is not clear why the present results from the Po Valley are compared to results from Leipzig made 7-8 years ago?**

(II) Author's response.
We revised the manuscript (Sect.4.1) to clarify that: (1) this comparison aims at reinforcing *the interpretation that the aerosol type represented by PC3* is the droplet mode aerosol; and (2) the correlation found between the droplet mode aerosol and the "brown" aerosol *demostrates that the "brown" aerosol is secondary in origin (the droplet mode is secondary in origin), and gives insights into its likely formation process.*

(III) Changes in manuscript.
We modified Sect. 4.1 to mention that to our knowledge, results from Leipzig are *the only work in literature showing a similar aerosol principal component to compare with.* The statistical methodology of the two works is the same, but datasets are very different, in time and space. Results from Leipzig were *based on five different statistical analysis based on different datasets* (two year data at eight concurrent measurement sites were available to create these five different dataset to be analysed). *Results were correlated to meteorological and air quality data.* Our work here was based on a shorter time period, and one measurement site. We do believe that results from Leipzig are statistically strong. These identified clearly the *PC representing the droplet mode aerosol*, which

310  shows *broad similarities with PC3* obtained in our work. *This allows to deduce with a reasonable statistical accuracy that PC3 does represent the droplet mode aerosol.*

(I) Comment from Referee:
**The authors have missed some other relevant work that also shows associations between ambient SOA and BrC - see for example X. Zhang et al. (2011; 2013).**

315  (II) Author's response.
The referee is right for mentioning that these are important works, both supporting our findings. In fact, we referenced the work by Zhang et al. (2013) in the former manuscript (P2L12, P10L13-15). In the revised manuscript, we referenced these works more explicitly.

(III) Changes in manuscript.
320  (*) In Sect.1 (P2L12-13), we indicated that *secondary organic aerosol (SOA) formed in the atmosphere contributes to the light absorbing carbon, as well (Moise et al., 2015), but only a few works have analysed secondary brown carbon associated to SOA (Zhang et al., 2013, 2011), and Saleh et al.(2013).*

(*) In Sect.5.3 (P10L13-15), we mentioned that *the composition found for this "brown" aerosol*
325  *(high OA, nitrates being a likely component), its formation process (involving aqueous phase reactions), and AAEs values, are all coherent with previous studies, which showed increased light absorption towards UV for SOA particles (Jacobson, 1999; Lee et al., 2013; Song et al., 2013; Zhang et al., 2013; Powelson et al., 2014; Lin et al., 2014; Laskin et al., 2015), and sources, composition, and AAE of light-absorbing soluble organic aerosol in urban areas (Zhang et al., 2013, 2011).*

330  (I) Comment from Referee:
**Section 5.2 and Figure 7: although the "paradigm" discussed here may have been developed in a prior paper, it is not something I think most readers will be familiar with (this reviewer was not). Provide the necessary explanation and context to interpret the present results.**

(II) Author's response.
335  We agree with the comment. In the revised manuscript, we added in Sect.5.2: (1) the context (including references to previous relevant works which have analysed the topic); (2) additional explanations to interpret results (including the revised Figure 7), and (3) a Table summarising these results (in particular, $k_{530}$, which is compared to relevant values of bulk aged OA in literature).

(III) Changes in manuscript.
340  (*) We indicated that *cluster analysis of aerosol spectral optical properties is becoming more and more used to infer information on aerosol type from optical data.* We mentioned that *Costabile et al. (2013) assessed spectral optical properties of key aerosol populations through Mie theory: soot, biomass burning, two types of organics, dust and marine particles were simulated through a sectional approach where each of these aerosol types was given a monomodal PNSD and a set*
345  *of three refractive indices (RIs) in the visible range. Relevant Angstrom Exponents of extinction, scattering, and absorption (EAE, SAE, AAE), SSA and its spectral variation (dSSA) were calculated. It was proved that these aerosol types separately cluster within a "paradigm" where SAE is on the y-axis, dSSA times AAE is on the x-axis, and SSA is on the z-axis.*

(*) We indicated that *experimental data of the "brown" aerosol do cluster in this paradigm* (Fig.
350  7), and that *the cluster of "brown" aerosol data is separated from all other simulated aerosol types, except that named "large organics". Data of "large organics" and "brown" aerosol do overlap, indicating that they may represent the same aerosol type. In fact, microphysical properties of the aerosol type named "large organics" were simulated to be same as those of the droplet mode aerosol (i.e., PNSD peaking in the "large" accumulation mode, 300-800 nm size range). Spectral optical prop-*
355  *erties of this "large organics" aerosol type were simulated by RIs of spectrally absorbing organic material in the visible region: these RIs, given the broad similarities in Fig.7, can be assumed to be those of the "brown" aerosol.*

(\*) We added Table 1 summarizing these results, and pointed that *this comparison ultimately allows to infer spectral optical properties of "brown" aerosol* (in particular, $k_{530}$, compared to relevant values recently reviewed by Lu et al. (2015) ).

(I) Comment from Referee:
**- Section 5.2, lines 28-34: perhaps this is in line with the above comment, but I was completely confused by this entire passage.**
(II) Author's response.
Text was changed (see previous comment).

(I) Comment from Referee:
**In my opinion, Figure 8 does not much at all to the paper ? I would recommend removing it.**
(II) Author's response.
We agree with the referee that this figure is not needed. However, it may be useful to connect results from different aerosol communities. As well, it reinforce findings. We would therefore prefer to keep it.

(I) Comment from Referee:
**Section 6: the findings do not "prove" the formation of BrC in the atmosphere.**
(II) Author's response.
We agree with the referee, and revised the text (P6L22-23) to mention that *findings show that there is "brown" aerosol in the atmosphere*.

(I) Comment from Referee:
**Pg. 7, line 21-22: "The dependence on the nitrate mass fraction (fNO3 , Fig.3d) is not obvious, as high AAE values and droplet mode scores are observed for both fNO3 <0.05 and fNO3 '0.25." This does not seem consistent with the discussion of nitrate's importance in the abstract or in Section 6.**
(II) Author's response.
The referee is right and we modified text accordingly, indicating (P1L4-5), that *findings show that "brown" aerosol... contains large concentrations of organic aerosol (OA) in droplet mode particles* and that *Nitrate is an additional likely component.*

**3 Figures and Table changes in manuscript.**

We added the following table to summarize spectral optical properties of brown aerosol, and improve clarity of Par. 5.2:

**Table 1.** Spectral optical properties of the "brown" aerosol in the visible range: AAE, SSA, and SAE (expressed as mean±standard deviation and variation range [minimum- maximum value]), and real and imaginary part ($k_\lambda$) of the complex refractive index (RI [$\lambda$]).

| $AAE_{467-660}$ | $SSA_{530}$ | $SAE_{467-660}$ | $RI_\lambda$ real part | $k_\lambda$ | [$\lambda$] |
|---|---|---|---|---|---|
| 3.5±0.8 [2.5-6] | 0.97±0.01 [0.92-0.99] | 0.8±0.3 [0-2] | 1.460 | $1.2 \cdot 10^{-2}$ | [467nm] |
| | | | 1.454 | $8 \cdot 10^{-3}$ | [530nm] |
| | | | 1.512 | $7.5 \cdot 10^{-3}$ | [660nm] |

Figure 4 was revised to include relevant uncertainties (not shown before):

[Figure]

**Figure 1.** Revised Figure 4 of the manuscript. *Relation between "brown" aerosol and Black Carbon (BC) to Organic Aerosol (OA) ratio (BC-to-OA). Absorption Angstrom Exponent at 467-660 nm (AAE) is plotted against BC-to-OA. Data color is the score of the droplet mode PC extracted by the statistical analysis. Data size is the median diameter of the particle surface size distribution ($d_{med(S)}$). Data indicated by "\*" show case-study values illustrated in Fig. 5. Grey bars indicate measurement uncertainty (Sect.2, and 3.1).*

390    Figure 6 was revised to include Shinozuka et al. (2009) and Lu et al. (2015) data:

[Figure]

**Figure 2.** Revised Figure 6 of the manuscript. *Dependence of Absorption Angstrom Exponent at 467-660 nm (AAE) on (a) organic aerosol mass fraction ($f_{OA}$), (b) Organic Aerosol (OA) to Black Carbon (BC) ratio (OA-to-BC), and (c) BC-to-OA ratio. Data color is Single Scattering Albedo at 530 nm. Data size is median diameter of particle surface size distributions ($d_{med_S}$, ranging from 50 to 300 nm). Data indicated by "\*" show case study values illustrated in Fig. 5. Relevant Pearson's correlation coefficients (r) are indicated. For comparison, we show previous results: (a) by Russell et al. (2010) (black line) and ? (grey lines, for the SSA bins of 0.90-0.92, 0.96-0.98 and 0.98-1.00) ; (c) by ? (black line), and Lu et al. (2015) (grey line). Note that AAE includes contributions from both BC and OA, whereas in panel (c), Lu et al. (2015) ?'s results refer to $AAE_{OA}$ only.*

Figure 7 was revised to improve clarity. The rev.Fig.7 includes panel a only, whereas panel b of the former Fig.7 was added in the Supplementary material:

[Figure]

**Figure 3.** Revised Figure 7 of the manuscript (including panel a of the former Figure 7 only).

[Figure]

**Figure 4.** Supplementary Figure, showing panel b of the former Figure 7.

The following figure was added in the supplementary material to show relative proportions of BC and OA in the AAE against BC-to-OA relation:

[Figure]

**Figure 5.** Supplementary Figure. *Relative proportions of BC and OA in the AAE vs BC-to-OA relation. BC-to-OA ratio (x-axis) against Absorption Angstrom Exponent (y-axis). Data points are sized by BC mass fraction ($f_{BC}$), and colored by organic aerosol mass concentration (OA). Vertical and horizontal bars show uncertainties (see Sect.3.1)*

395    The following figure was added in the supplementary material to show $AAR_{BC,OA}$ against $AAE_{BC}$:

[Figure]

**Figure 6.** Supplementary Figure. *Dependance of the Absorption Angstrom Exponent (AAE) due to organic aerosol (OA) from the BC-to-OA ratio. The AAE due to BC and OA ($AAE_{BC,OA}$) of the dust-free bulk AAE is indicated by black squares. (A subset of data is indicated with uncertainty-bars (grey lines), the interpolating curve of the whole dataset being indicated by the black curve). $AAE_{BC}$ used to derive BC is also indicated (grey squares).*

The following figure was added in the supplementary material to show loadings and scores calculated by PCA of the four PCs of PNSD:

[Figure]

**Figure 7.** Supplementary Figure. *Principal components (PC1-PC4) of the Particle Number Size Distributions: (a-d) weekly diurnal cycles of scores (red, Week Day; green, Week End; black, all data), and (e) loadings.*

The following figure was added in the supplementary material to show extensive spectral optical properties during the two field campaigns:

[Figure]

**Figure 8.** Supplementary Figure. *Spectral optical properties during field campaigns: y-axis shows data of scattering and absorption coefficients, colored by the Scattering Angstrom Exponent (SAE), and Absorption Angstrom Exponent (AAE), respectively.*

**400 References**

[revised manuscript text omitted]

---

## Referee Report (RR1)

The authors' effort to address the comments are acknowledged and appreciated. However, I still believe that my two major concerns from the previous review still hold. I think it is best to make my points by doing the following simplified calculation.

Assuming no effect of internal mixing (just to simplify the calculation), one can calculate MAC of the aerosol (BC + OA) as a weighted average of the MACs of the components:

$$MAC_{BC+OA} = MAC_{BC}\ C_{BC} + MAC_{OA}\ C_{OA} \tag{1}$$

Where $C_{BC}$ and $C_{OA}$ are the concentrations. This can be written in terms of BC-to-OA ratio as:

$$MAC_{BC+OA} = MAC_{BC}\ (1 + BC\text{-to-}OA) + MAC_{OA}\left(1 + \frac{1}{BC\text{-to-}OA}\right) \tag{2}$$

And AAE is:

$$AAE_{BC+OA} = \frac{d\ln(MAC_{BC+OA})}{d\ln(\lambda)} \tag{3}$$

Solving equations (1-3), one can mathematically reproduce the data in Figure 4 and SI Figure 1 (Figure 6 in the authors' response).

The blue curve in Figure R1 was calculated using $MAC_{BC}$ (532nm) = 10 m²/g, $MAC_{OA}$ (532nm) = 0.2 m²/g [just an assumption based on general values in the literature. You can use different values, but won't change the general picture], $AAE_{BC}$ = 1.1 [the value assumed by the authors], and $AAE_{OA}$ = 6 [chosen to reproduce measurements at low BC-to-OA].

The red curve was calculated using the same values, with the only difference being $AAE_{BC}$ = 1.8. Let's put the difference between the blue and red curves aside for now. Looking at any of the two curves, one can see that $AAE_{BC+OA}$ increases with decreasing BC-to-OA ratio in a fashion very similar to what the authors report, simply due to the decreased contribution of BC to AAE. In other words, a constant $AAE_{OA}$ can explain the data (whether that OA is primary or secondary does not matter). Therefore, the authors' conclusion that "brown" aerosol is exclusively secondary OA (e.g. line 296 and line 455) does not necessarily follow from the data. The correlation with f44 and the droplet mode is not enough. All they can say is that aged OA contributes to the brown aerosol, but they cannot say that the brown aerosol is exclusively secondary (unless they show that all OA is secondary, which I don't think is the case).

Now to the second major point concerning $AAE_{BC}$. In the previous review, I made the point that $AAE_{BC+OA}$ should converge to $AAE_{BC}$ at large BC-to-OA ratios. This is supported by the calculations shown in Figure R1. The authors assume $AAE_{BC}$ of 1.1 in their analysis, while their data (Figure 4 and SI Figure 1) clearly show that AAE plateaus at ~1.8 at large BC-to-OA ratios.

They explained this discrepancy in the revised manuscript as "*due to any spectrally light absorbing material that the AMS could not detect (refractory material, or material in particles smaller than 100 nm and larger than 1 _m).*"

This is not convincing. First, what is the light-absorbing refractory material with such a high AAE (it needs to be >> 1 in order to have such a big influence)? It could be dust, but the authors say that they exclude data that had contribution from dust.

Second, let's assume that the contribution is from OA particles that the AMS could not see (too small or too large particles). That would mean the AMS missed A LOT of OA mass. This can be explained by looking at the difference between the blue and red curves in Figure R1. The red curve is very similar to the authors' data (e.g. Figure 4 in the manuscript). AAE plateaus at 1.8. If $AAE_{BC}$ is 1.1, that would mean what the authors report as BC-to-OA = 20, should actually be 0.5 (where the dashed black line intersects the blue curve in Figure R1). Of course, this calculation is simplified, but the point is that the BC-to-OA has to be grossly underestimated (at least an order of magnitude) for the authors' explanation to hold.

I don't think this is the case. The more logical explanation is that the AAE measurements, for some reason, are overestimated by ~80%. And as I pointed out in the previous review, this would explain the unusually large $AAE_{BC+OA}$ reported in this study.

I think the authors should try to address this bias, or at the very least clearly state it in the manuscript and discuss the implications.

Finally, it is not clear why the authors define "brown" aerosol as something different than brown carbon. Do they mean that there are non-organic (non-dust) components that are also brown? If yes, they need to justify. If not, it seems to me that brown aerosol and brown carbon are synonymous.

[Figure]

Figure R1

---

## Referee Report (RR2)

Review of revised manuscript submitted by Costabile et al.: "Characteristics of "brown" aerosol in the urban Po Valley atmosphere"

The revised manuscript addressed some of the referee comments from the first review, but significant problems remain (including new problems that have emerged in this version). Thus, even after the revision, I would still rate the manuscript as needing major revisions. My major comments are:

1. The discussion about the source of the "brown" aerosol is completely convoluted. For example, in the abstract and conclusion, the authors state "it does not necessarily equate to brown carbon". However, most of the manuscript presents support for the hypothesis that this material is indeed BrC.

2. The influence of the case study seems to drive a lot of the broad conclusions (for example, the trends in Figure 3). If this one day (out of 40 total included in the analysis) is removed, to the broad trends hold up? How would the values in Table 1 be different if the one day case study is removed?

3. I also have some problems with the authors' interpretations of their data – especially the data presented in Table 1. Specifically, the authors suggest that PC1, PC2 and PC4 are "BC primary aerosol" (line 274). The support for this statement is that "all are correlated to BC and $f_{BC}$", but I completely disagree with that assessment. PC2 and PC4 are not correlated at all with BC or $f_{BC}$ – the highest $R^2$ value is 0.04 for the correlation between PC2 and $f_{BC}$. PC1 is not correlated with $f_{BC}$, and is only weakly correlated with BC ($R^2$ value is only 0.36). Further, line 300 claims "a robust statistical relation linking AAE, this "droplet" mode component (PC3), and OA-to-BC, together with $f_{OA}$, $d_{med(S)}$, $f_{44}$ and $f_{43}$." I do not think that Table 1 provides evidence for a robust statistical relationship, given that the highest $R^2$ value among all the relationships analyzed was $< 0.5$.

4. Building on comment #3 above, since PC3 represents larger particles compared with PC1, PC2, and PC4, it is not at all surprising that it is more strongly associated with aerosol optical properties, since it represents particles that are more optically active. This seems relevant in the data interpretation (put another way, of course PC4 has no correlation with aerosol optical properties – since these particles are so small).

5. Again, in the abstract and conclusion, it is stated that nitrate likely contributes to the aerosol absorption (lines 9 and 473). However, I see no evidence from the data to support these statements. In fact, the authors seem to acknowledge this (discussion on line 310).

6. In the abstract and conclusion, the authors claim that theirs is the first study to "consider these issues" – such a claim needs to be clarified, as many prior studies have looked into secondary BrC.

7. I made this same comment in the first review, but I really don't understand why the prior results from Costabile et al. (2009) have been added to Figure 1? In the text (lines 289-295), the authors suggest that this study was the only other instance in the literature where a droplet

mode PC was identified from size distribution measurements. That may be because aerosol size distribution data are not routinely subjected to this PC analysis, but the observation of a droplet mode is NOT unique to Costabile et al. (2009). Overall, Figure 1 and this discussion are quite confusing.

8. I am still not sure how Figures 7 and 8 (and their associated discussion) add to the manuscript. If anything, these figures and discussion add confusion.

9. Finally, the quality of the writing needs to be improved throughout the manuscript. There are too many grammatical corrections for this referee to itemize.

---

## Author Response (AR2)

**LETTER TO EDITOR**

Dear Editor,

we would like to thank you and both reviewers for the insightful comments. To answer these comments, the manuscript has changed substantially (detailed point-by-point answers to reviewers's comments are reported below). The main changes from the older version are as follows.

First, with the criticisms raised we realized there was an error in the dataset of the original manuscript (indeed, reviewer 1 was right in noting that there was a significant underestimation of OA). We do apologize for the error in the data of the original manuscript. We realized that the corrections following this mistake are substantial and significant, but the revised version shows a better consistency with the literature, in comparison with the original draft. In particular, the relation between AAEs and BC-to-OA ratios (rev. Fig.5) is now very similar to that found by both Lu et al.(2015) and Saleh et al. (2014).

Second, OA properties during the winter field campaign are now discussed in detail in [1]. In [1] we proved the correspondence between larger AAE values and larger values of an Oxygenated Organic Aerosol (OOA) component originating from biomass burning and influenced by aqueous phase processing (aqSOA). We apologize for the confusion about this point in the original manuscript. In the revised manuscript any discussion about SOA was deleted, the reader being referred to [1] for the demonstration of the correspondence between BrC and aqSOA.

Finally, to answer comments raised by both reviewers, the way aerosol size and refractive index influence BrC is analysed in detail in the revised manuscript (rev. Fig.1 and relevant text). We matched AAE measurements to numerical simulations (Mie theory) of  $AAE(d_p, m, \lambda)$  resolved by particle diameter ( $d_p$ ) and wavelength ( $\lambda$ ) dependent complex refractive index ( $m(\lambda)=n(\lambda)-ik(\lambda)$ ). BrC patterns (measured) were found to match those theoretically expected for BrC with  $m(\lambda)$  taken from the literature. This consistency has allowed us to gain more strength and confidence from measurements, and to identify and characterize BrC in a clearer fashion, in comparison to the original draft.

Relevant Figures and Tables (and relevant text) have changed from the older revision as follows: (i) three figures (old Fig.1, 7 and 8) and one Table (Tab.2) deleted as requested by reviewer 2; (ii) one figure modified (older Fig.2, now Fig.1) to show measurements and simulations of AAE against aerosol size (instead of AAE vs droplet mode aerosol score); (iii) four figures (older Fig.3,4,5,6, now Fig.2,3,4,5) and one table (Tab.1) corrected (due to the error in the dataset) and slightly modified.

The review process has significantly changed (to our opinion improved) the manuscript. Relevant conclusions did not change, but are now more consistent to literature and theory. We wish this last version of the manuscript can be accepted to finalize ACP review purposes.

Sincerely yours,

Francesca Costabile (on behalf of all of the co-authors)

**REFERENCES**

 Gilardoni, S., Massoli, P., Paglione, M., Giulianelli, L., Carbone, C., Rinaldi, M., Decesari, S., Sandrini, S., Costabile, F., Gobbi, G.P., Pietrogrande, M.C., Visentin, M., Scotto, F., Fuzzi, S., Facchini, M.C.: Direct observation of aqueous secondary organic aerosol from biomass burning emissions, P. Natl. A. Sci., 113.36: 10013-10018, 2016.

**AUTHOR'S ANSWERS TO REVIEWERS'S COMMENTS**

We thank both reviewers for their insightful comments, and also for insisting on certain points. As already mentioned in the letter to Editor, to answer these points we realised there was an error in the original dataset. We wish to apologise for this error to both reviewers. Following correction of the error, several points shall be already answered.

Our detailed point-by-point answers to reviewer's comments are reported below (in Bold, reviewer's comments (RC); in plain text, author's answers (AA)).

**1 Referee 1**

RC) Looking at any of the two curves, one can see that  $AAE_{BC,OA}$  increases with decreasing BC-to-OA ratio in a fashion very similar to what the authors report, simply due to the decreased contribution of BC to AAE. In other words, a constant  $AAE_{OA}$  can explain the data (whether that OA is primary or secondary does not matter). Therefore, the authors conclusion that brown aerosol is exclusively secondary OA (e.g. line 296 and line 455) does not necessarily follow from the data. The correlation with  $f_{44}$  and the droplet mode is not enough. All they can say is that aged OA contributes to the brown aerosol, but they cannot say that the brown aerosol is exclusively secondary (unless they show that all OA is secondary, which I dont think is the case).

AA) The reviewer correctly notes that this major finding does not follow from this paper. It is indeed the subject of another paper just published [1], in which we prove that the larger AAE values correspond to larger values of an Oxygenated Organic Aerosol (OOA) component originating from biomass burning and influenced by aqueous phase processing (referred to as aqSOA).

We apologize for the confusion in the original manuscript. In the revised manuscript, we explain the increase of AAE, and thus BrC formation, with the formation of secondary organic aerosol in the aqueous phase, as proved in a different paper [1]. Any discussion concerning secondary origin was deleted, the reader being referred to [1] for a complete characterization of OA properties.

RC) Now to the second major point concerning  $AAE_{BC}$ . In the previous review, I made the point that  $AAE_{BC,OA}$  should converge to  $AAE_{BC}$  at large BC-to-OA ratios. This is supported by the calculations shown in Figure R1. The authors assume  $AAE_{BC}$ of 1.1 in their analysis, while their data (Figure 4 and SI Figure 1) clearly show that AAE plateaus at 1.8 at large BC-to-OA ratios. They explained this discrepancy in the revised manuscript as due to any spectrally light absorbing material that the AMS could not detect (refractory material, or material in particles smaller than 100 nm and larger than 1  $\mu$  m). This is not convincing. First, what is the light-absorbing refractory material with such a high AAE (it needs to be >> 1 in order to have such a big influence)? It could be dust, but the authors say that they exclude data that had contribution from dust. Second, lets assume that the contribution is from OA particles that the AMS could not see (too small or too large particles). That would mean the AMS missed A LOT of OA mass. This can be explained by looking at the difference between the blue and red curves in Figure R1. The red curve is very similar to the authors data (e.g. Figure 4 in the manuscript). AAE plateaus at 1.8. If  $AAE_{BC}$  is 1.1, that would mean what the authors report as BC-to-OA = 20, should actually be 0.5 (where the dashed black line intersects the blue curve in Figure R1). Of course, this calculation is simplified, but the point is that the BC-to-OA has to be grossly underestimated (at least an order of magnitude) for the authors explanation to hold. I dont think this is the case. The more logical explanation

is that the AAE measurements, for some reason, are overestimated by ~ 80%. And as I pointed out in the previous review, this would explain the unusually large  $AAE_{BC,OA}$  reported in this study. I think the authors should try to address this bias, or at the very least clearly state it in the manuscript and discuss the implications.

AA) The reviewer is correct.

There was an error in the dataset. The OA-to-BC ratios corrected are more than double that uncorrected, and the BC-to-OA ratios revised are lower than 1 instead of 10 (see rev. Fig. 2, 3, and 5). Indeed the reviewer was right in noting that that there was a significant underestimation of OA. We apologize for the error, and thank the reviewer for insisting on the importance of this point. After correction of the error, the relation between AAE and BC-to-OA is now far more consistent to literature than what found in the original manuscript (see revised Fig.3 and Fig.5).

RC) Finally, it is not clear why the authors define brown aerosol as something different than brown carbon. Do they mean that there are non-organic (non-dust) components that are also brown? If yes, they need to justify. If not, it seems to me that brown aerosol and brown carbon are synonymous.

AA) We apologize for the confusion about the ambiguous use of the term brown aerosol in the original manuscript. In the revised version we use the term brown carbon instead.

**2 Referee 2**

RC) 1. The discussion about the source of the brown aerosol is completely convoluted. For example, in the abstract and conclusion, the authors state it does not necessarily equate to brown carbon. However, most of the manuscript presents support for the hypothesis that this material is indeed BrC.

AA) We apologize for the confusion about the ambiguous use of the term brown aerosol in the original manuscript. In the revised version we use the term brown carbon instead.

RC) 2. The influence of the case study seems to drive a lot of the broad conclusions (for example, the trends in Figure 3). If this one day (out of 40 total included in the analysis) is removed, to the broad trends hold up? How would the values in Table 1 be different if the one day case study is removed?

AA) In the revised manuscript we clarify that the case study does not last one day, but 1.5 h, and show case study datapoints in all the relevant figures (Fig.2, 3, and 5) to illustrate how these few data relate to the broad trends (we removed accordingly panel f of Fig.5 because AAE case study data are indicated in all figures).

**RC) 3. I also have some problems with the authors interpretations of their data - especially the data presented in Table 1.**

AA) The reviewer is correct. We apologize for the confusion of this part of the original manuscript.

In the revised manuscript Table 1 was modified. First, we realized that there was an error in the dataset, and modified Table 1 accordingly. Then, we decided to show in Table 1 only statistically significant correlations (p<0.001). In Table 2 of the supplementary material we show relevant statistical significance, i.e. the matrix of Bonferroni Probabilities (p) associated to Pearson's correlation coefficients (r)).

RC) (continued) Specifically, the authors suggest that PC1, PC2 and PC4 are BC primary aerosol (line 274). The support for this statement is that all are correlated to BC and fBC, but I completely disagree with that assessment. PC2 and PC4 are not correlated at all with BC or fBC the highest R2 value is 0.04 for the correlation between PC2 and fBC. PC1 is not correlated with fBC, and is only weakly correlated with BC (R2 value is only 0.36).

AA) In the revised manuscript PCs interpretation is discussed in more detail (Par.3.3, and Par.4.1). We added numerical simulations of  $AAE(d_p, n, \lambda)$  (see rev. Fig.1) to show the way PC1, PC2, and PC3 relate to theoretical values expected for BC and BrC. In addition, we showed the statistical significance of correlations (see above), and added values of correlation in the Winter and the Fall (Table 1 of the supplementary material). Finally, we decided to exclude PC4 from the analysis. PC4 is only a smaller component (approx. 8% of the variance) with loadings peaking in the nucleation mode size range (20-40 nm): as indicated by this reviewer in comment n.4, PC4 is not expected to affect significantly aerosol optical properties.

Therefore, in the revised manuscript PC1 represents the aerosol originating in the traffic rush hour in the urban area, due to local emissions enriched in OA (r=0.9, p<0.001 in the Winter), in agreement with the literature (e.g., Costabile et al., 2009; Brines et al., 2015, and references therein). PC2 represents the nocturnal urban aerosol related to residential wood burning emissions (r=0.4, p<0.001, with both  $f_{BC}$  and  $f_{OA}$  in the Winter).

RC) (continued) Further, line 300 claims a robust statistical relation linking AAE, this "droplet" mode component (PC3), and OA-to-BC, together with fOA, dmed(S), f44 and f43. I do not think that Table 1 provides evidence for a robust statistical relationship, given that the highest R2 value among all the relationships analyzed was < 0.5.

AA)As mentioned before, in the revised manuscript Tab.1 was modified. We identified statistically significant correlations (values of Bonferroni Probabilities p shown in Tab.2 of the supplementary material). These indicate that correlations linking AAE to the droplet mode aerosol (r=0.66 with r=0.63 in the Fall, r=0.67 in the Winter, and p<0.001), AAE to OA-to-BC (r=0.78 with r=0.55 in the Fall, 0.82 in the Winter, and p<0.001), AAE to  $d_{med(S)}$  (r=0.60 in the Fall, 0.71 in the Winter, and p<0.001), and the droplet mode aerosol to OA-to-BC (0.54, p<0.001) are all statistically significant correlations. In addition, in the revised manuscript the relation linking these variables is analysed in more detail through numerical simulations of  $AAE(d_p, \lambda, m_\lambda)$ . We show that measurements (underlying these correlations) match relevant patterns theoretically expected for BrC (rev. Par.4.1, and rev. Fig.1 and Fig.3).

With regard to  $f_{43}$  and  $f_{44}$ , we decided to delete relevant correlations from Table 1 since we proved in [1] the correspondence between larger AAE and aqSOA.

RC) 4. Building on comment 3 above, since PC3 represents larger particles compared with PC1, PC2, and PC4, it is not at all surprising that it is more strongly associated with aerosol optical properties, since it represents particles that are more optically active. This seems relevant in the data interpretation (put another way, of course PC4 has no correlation with aerosol optical properties since these particles are so small).

AA) The reviewer is right in noting that this part of the original manuscript was not clear enough.

In the revised manuscript (Par.4.1 and Fig.1) we show that AAE values increase with increasing the optically relevant aerosol size  $(d_{med(S)})$ , as suggested by the reviewer. As well, through numerical simulation (Mie theory) of AAE $(d_p, n, \lambda)$  expected for BrC, we demonstrate that the large AAE measured for the BrC droplet mode aerosol can be explained only by the coupling of peculiar  $m_{(\lambda)}$  values and larger particle size.

This is a major finding because characterises BrC: it emphasizes that BrC properties depend on both aerosol chemical composition  $(m_{(\lambda)})$  and size distribution.

**RC) 5. Again, in the abstract and conclusion, it is stated that nitrate likely contributes to the aerosol absorption (lines 9 and 473). However, I see no evidence from the data to support these statements. In fact, the authors seem to acknowledge this (discussion on line 310).**

AA) The reviewer is correct. As mentioned before, there was an error in the dataset. We do apologize for this error and do thank the reviewer for insisting on this point. Following its correction, aerosol fractions do change (see revised Fig.2). There is now evidence that the larger AAE values correspond to larger nitrate mass fraction. In the revised manuscript, we mentioned this correspondence suggesting that nitrate is likely associated associated with BrC particles.

**RC) 6. In the abstract and conclusion, the authors claim that theirs is the first study to consider these issues. Such a claim needs to be clarified, as many prior studies have looked into secondary BrC.**

AA) The reviewer is correct.

That claim was too general. In the revised manuscript, we clarified that this study provides experimental evidence that the size distribution of BrC associated with the formation of secondary aerosol is dominated by the droplet mode, consistent with recent findings pointing to the role of aqueous reactions within aerosol particles in the formation of BrC.

RC) 7. I made this same comment in the first review, but I really don't understand why the prior results from Costabile et al. (2009) have been added to Figure 1? In the text (lines 289-295), the authors suggest that this study was the only other instance in the literature where a droplet mode PC was identified from size distribution measurements. That may be because aerosol size distribution data are not routinely subjected to this PC analysis, but the observation of a droplet mode is NOT unique to Costabile et al. (2009). Overall, Figure 1 and this discussion are quite confusing.

AA) To avoid possible confusion, Figure 1, and the associated discussion, were deleted from the revised manuscript .

RC) 8. I am still not sure how Figures 7 and 8 (and their associated discussion) add to the manuscript. If anything, these figures and discussion add confusion.

AA) To avoid possible confusion, both Figures (and their associated discussion) were removed from the revised manuscript.

RC) 9. Finally, the quality of the writing needs to be improved throughout the manuscript. There are too many grammatical corrections for this referee to itemize.

AA) A mother-tongue collaborator checked the grammatics of the revised manuscript.

Manuscript prepared for Atmos. Chem. Phys. with version 2014/09/16 7.15 Copernicus papers of the LATEX class copernicus.cls. Date: 18 October 2016

**Characteristics of "brown" aerosol Brown Carbon in the urban Po Valley atmosphere**

F. Costabile1, S. Gilardoni2, F. Barnaba1, A. Di Ianni1, L. Di Liberto1, D. Dionisi1, M. Manigrasso3, M. Paglione2, V. Poluzzi4, M. Rinaldi2, M.C. Facchini2, and G. P. Gobbi1

[revised manuscript text omitted]

where χ is the shape factor, Cc(dm) and Cc(da) are the slip correction factors based on dm and da respectively, ρp is the particle density, and ρ0 is the unit density (1 g·cm-3). In applying Eq.2
to convert APS data, we assumed: particle diameter (dp) = dm represents the true particle diameter; Cc(dm) = 1 and Cc(da) = 1 (continuum regime); χ = 1 (spherical particles); ρp continuously varying from 1.6 to 2 g·cm-3.

Particle Number Size Distributions PNSDs (i.e.,  $n_N(log_{10}d_m) = \frac{dN}{dlog_{10}(d_m)}$ ) measured by SMPS and APS (PNSDSMPS, PNSDAPS) overlap for  $d_m$  ranging from 460 nm to 593 nm. In this size range, PNSDSMPS and PNSDAPS were replaced by PNSDfitted. PNSDfitted was assumed to vary according to a power-law function (Junge size distribution) (Khlystov et al., 2004; Seinfeld and

Pandis, 2006) (Eq.3):

275

$$n_N(\log_{10}d_m) = \frac{c}{d_m^{\alpha}},\tag{3}$$

The coefficients c and  $\alpha$  were calculated by an iterative procedure: (i) c was randomly initialized from 0 to 1000; (ii)  $\alpha$  was calculated by Eq.3 constraining values from 2 to 5, as typically found for atmospheric aerosols (Seinfeld and Pandis, 2006). PNSD*fitted* replaced PNSD*APS* and PNSD*SMPS* when their relative difference ( $\frac{\mathbf{d}_{T}\delta(PNSD)}{\mathbf{d}_{T}}$ , Eq.4):

$$\underline{d_r}\delta(\underline{PNSD}) = \frac{|PNSD_{SMPS} - PNSD_{APS}|}{max[PNSD_{SMPS}, PNSD_{APS}]}$$
(4)

was larger than 0.1 cm-3. This procedure was considered acceptable if: (i) the minimum mean squared error between  $PNSD_{fitted}$  and  $PNSD_{APS}$  was less than 1%; (ii) correlation coefficients between  $PNSD_{fitted}$  and  $PNSD_{SMPS}$ , and between  $PNSD_{fitted}$  and  $PNSD_{APS}$  were larger than 0.8 . A number of (98 of the records did not verify these conditions, and were checked by visual inspection: 94 of them were discarded, and 4 accepted). The final dataset contained PNSD data based on dm from 14.1 to 429.4 nm measured by the SMPS, from 446.1 to 699 nm generated by

290 the fitting procedure, and from 0.7 to 14  $\mu$ m measured by the APS. The Particle Surface Size Distribution (PSSD, i.e.  $n_S(log_{10}d_m) = \frac{dS}{dlog_{10}(d_m)}$ ) and Particle Volume Size Distribution (PVSD, i.e.  $n_V(log_{10}d_m) = \frac{dV}{dlog_{ro}(d_m)}$ ) were calculated from this PNSD under the hypotheses of spherical particles (Hinds, 1999; Seinfeld and Pandis, 2006).

**3.3 Principal component analysis of PNSD**

- 295 PNSDs were statistically analysed through Principal Component Analysis (PCA) PCA was calculated following the findings of a previous long-term study over multiple sites (8 concurrent stations) to identify key aerosol types with known modality. The relevant methodology was described in a previous study by Costabile et al. (2009). Four principal components (PC1-PC4) were extracted, explaining 900f the variance. We interpreted these PCs based on: (i) their statistical properties, i.e.
- 300 "scores" and "loadings" (loadings indicate correlations between PNSDs and PCs, i.e. the "mode" of the PNSD associated to the PC); (ii) Pearson's correlation coefficients (r) shown in Table 1, between PCs, AAE, and PM1 mass fractions of BC, organics, nitrate, sulfate, and ammonium ( $f_{BC}$ ,  $f_{OA}$ ,  $f_{NO3}$ ,  $f_{SO4}$ ,  $f_{NH4}$ ). The median diameter of the particle surface size distribution ( $d_{med(S)}$ ) in Table 1 is intended to add information on optically relevant aerosol sizes. The OA to BC ratio (OA-to-BC) in
- Table 1 is intended to indicate both combustion characteristics (higher for biofuels than for fossil fuel combustion), and aerosol ageing (lower for fresh aerosols) (Saleh et al., 2014; Bond et al., 2013). Correlations to f43 and f44 (defined as the ratio of the AMS signal at m/z 43 and m/z 44, respectively, to the total organics AMS signal) in Table 1 are intended to add information on the oxidised OA. The higher the f44, the more oxidised the OA; the higher the f44/f43 ratio, the lower the volatility of this oxidised OA (??Moise et al., 2015).
- STO this oxidised OA (...Woise et al., 2015).

Statistical properties used to interpret PC are "scores" and "loadings". PCs-In short, Principal Components (PCs) retained in the analysis were arranged in decreasing order of variance explained ( $\varkappa_k$ , called eigenvalue of PCk), PC1 being the component explaining the largest  $\varkappa_k$ . The  $k^{th}$  eigenvector is composed of scalar coefficients describing the new PCk as a linear combination of the orig-

- 315 inal variables (the original variables are the time series of dN/dlog(dp)). Coefficient Factor loadings of PCk represent the relative weight (in terms of correlation) of dN/dlog(dp) variables in of the original variables in the PCk. Factor loadings of PCk represent these coefficients scaled 
[revised manuscript text omitted]
 OA-to-BC, together with fOA, dmed(S), f44 and f43. Figure ?? shows this relation. When PC2) match these patterns. Larger AAEs  $(3.2\pm0.9)$  correspond to the droplet mode PC scores positive (PC scores >0 indicate that this aerosol type forms), the aerosol (PC3), and are
- 435 similar to the AAE expected for Brown Carbon aerosol. Fig.1 suggests that the threshold value of

AAE467-660 distinguishing BrC is AAEis greater than 2.5. Both AAE and droplet mode increase with increasing OA-to-BC, and  $f_{44}$ , both indicators of OA aged in the atmosphere (?Bond et al., 2013). Data, therefore, identify a "brown" acrosol type, >2.3. The dotted black line in Fig.1 shows the increase of the AAE measured with increasing  $d_{med(S)}$ ; brown and grey lines show the increase

- of the AAE theoretically expected with increasing aerosol size,  $m_{(\lambda)}$  being constant. Comparing these three lines, it is evident that peculiar values of  $m_{(\lambda)}$  are necessary in addition to the larger particle size to match the large AAE measured for the droplet mode BrC aerosol. These are  $k_{(467)}$ = 0.026±0.001,  $k_{(530)}$  = 0.017±0.001,  $k_{(660)}$  = 0.014±0.001, and  $n_{467}$ =1.47±0.01 (Sect.3.4). This finding emphasizes that BrC properties depend on both aerosol size distribution and chemical composition
- 445 (i.e.an aerosol showing AAE from 2.5 to 6, having high "droplet" mode PC scores. The dependence of this "brown" aerosol on ,  $\underline{m}_{(\lambda)}$ ).

Figure 2 investigates the link between chemical, microphysical, and optical properties. AAE is plotted against  $PM_1$  major costituents, and relevant ratios, is illustrated in Fig. 2 and Fig. 3, where the brown aerosol formation is indicated by the combined increase of AAE (y-axis) and droplet mode PC

- 450 chemical components (OA, BC, nitrate, and sulfates PM1 mass fractions,  $f_x$ ). Grey markers show the longest available time series for AAE and  $f_x$ , while coloured markers show the (shorter) time series including the droplet mode aerosol score (marker color). This "brown" aerosol formation depends on the organic mass fraction ( $f_{OA}$ , Fig.2a), and is inversely correlated to the mass fractions of BC ( $f_{BC}$ , Fig.2b), and sulfate ( $f_{SO4}$ , Fig.2c). The dependence on the nitrate mass fraction ( $f_{NO3}$ , Fig.2d) is not
- 455 obvious, as high AAEvalues colour) and droplet mode scores are observed for both  $f_{NO_3} < d_{med(S)}$ (marker size) (see Sect.3). BrC particles (i.e, particles with AAE>2.3) shows higher  $f_{NO3}$  and lower  $f_{BC}$  values coupled to larger  $d_{med(S)}$  and high droplet mode aerosol scores. Average values corresponding to BrC aerosol population are as follows (mean±standard deviation): AAE=3.2±0.9,  $f_{NO3}=0.38\pm0.05$ ,  $f_{BC}=0.01\pm0.01$ ,  $f_{OA}=0.35\pm0.04$  and  $f_{NO_3}\simeq0.25_{SO4}=0.1\pm0.02$ , and  $SSA_{530}=0.98\pm0.01$

460 (and  $\sigma_{a467} = 7.6 \pm 3.33 Mm^{-1}$ ,  $\sigma_{s467} = 312 \pm 64 Mm^{-1}$ , and SAE=0.5±0.4). The relation between this "brown" aerosol and the BC-to-OA ratio (or its inverse OA-to-BC ratio = Both Fig.2 and Tab.1 show that there is no direct correlation between AAE and  $f_{QA}$ . AAE correlates with larger particles (larger  $d_{med(S)}$ ) in the droplet mode (larger PC3 scores), while the  $f_{QA}$  / $f_{BC}$ )is shown in Fig.3.Uncertainties of AAE , and BC-to-OA in Fig. 3 were calculated

- 465 propagating uncertainties of PSAP-derived  $\sigma_a$  (Seet.2.2), BC derived by the AAEBC attribution method (Sect.3.1), and AMS-derived OA (Sect.2.3). The area showing the brown aerosol (AAE from 2.5 to 6.6, positive droplet mode PC scores)has high OA-to-BC ratios (note correlation r = 0.78 correlates with smaller particles (lower  $d_{med(S)}$ ) from residential heating emissions (larger PC2 scores). There is, however, a significant correlation between AAE and OA-to-BC ratio, Table
- 470 the ratio of OA to BC (r=0.78, p<0.001). The inverse dependence between this "brown" aerosol formation and the BC-to-OA ratio confirms results of the statistical analysis (correlations with  $f_{44}$  and  $f_{43}$ ) indicating that the brown aerosol is an aged OA. Indeed, it is typical that BC contribution

declines, and OA contribution increases, as the smoke aerosol ages in the atmosphere. Relative proportions of BC and OA in the AAE vs-OA-to-BC ratio indicates either combustion characteristics

- 475 (higher for biofuels than for fossil fuel combustion) or aerosol ageing (lower for fresh aerosols) (Saleh et al., 2014; Bond et al., 2013). The relation between AAE and BC-to-OA relation are indicated in the supplementary material. It is shown that the increase in AAE with decreasing the ratios is illustrated in Figure 3 (light-grey markers show the longest available time series for AAE and BCto-OAratio is not simply due to the decreased contribution of BC. The AAE of the aerosol, while
- 480 coloured markers show the time series including the droplet mode aerosol score and the  $f_{NO3}$ ). 
[revised manuscript text omitted]
 ( $f_{BC}$ BC), organics ( $f_{OA}$ OA), nitrate ( $f_{NO3}$ NO3-), sulfate ( $f_{5O4}$ SO42-), and ammonium ( $f_{NH4}$ NH+4); optically relevant aerosol size representative of the entire aerosol population (calculated as median mobility diameter of the particle surface size distribution<del>(</del>,  $d_{med(S)}$ ); BC mass concentration (BC); organic aerosol (OA) to BC ratio (OA-to-BC ); ratio ratios. See Tab.1 and Tab.2 of the AMS signal at m/z 44 and m/z 43 to Supplementary material for all the total organics AMS signal ( $f_{44}$  and  $f_{43}$ )relevant values. Note that PC3 is the "droplet" mode PC.

| r                                                       | AAE                | $d_{med(S)}$                | BC                               | $\mathbf{f}_{BC}$     | OA                   | $f_{OA}$                     | OAtoBCOA-to-                  |
|---------------------------------------------------------|--------------------|-----------------------------|----------------------------------|-----------------------|-----------------------------|------------------------------|-------------------------------|
| AAE drived (S)                                          | 0.60               | <del>0.48-</del> 1          | - <del>0.26</del> - 0.2 4 | -0.37_0.38            | <del>0.56 _</del>           | <del>0.78 _0.68</del> | -0.31-0.37                    |
| AAE                                                     | <del>0.60-</del> 1 | <del>0.31 0.60</del> | <del>0.08</del> - 0.26    | <del>0.00</del> -0.52 | -0.1-0.40                   | -0.35-0.34                   | - <del>0.22 0.78</del> |
| PC3 PC1                                                 | <del>0.67 _</del>  | <del>0.60</del>             | -0.35 0.56                       | <del>-0.42</del>      | <del>0.64 0.83</del> | <del>0.60</del> –            | <del>-0.35</del>              |
| $d_{med(S)}$ PC2                                        | <del>0.48 –</del>  | -0.52                       | <del>0.22</del>                  | <del>-0.48</del> _    | <del>0.38</del> _           | <del>0.48.0.31</del>  | <del>-0.12</del>              |
| PC3                                                     | 0.66               | 0.60                        | real part -0.38                  | $k_{\lambda}$ -0.53   | →0.12                | -0.52                        | 0.54                          |
| $1512 \pm 0.0175 \pm 10^{-3} \pm 0.001660$ m h si - h t |                    |                             |                                  |                       |                             |                              |                               |

 $1.512\pm0.017.5 \cdot 10^{-3}\pm0.001660$ nmheight

**925 FIGURES**

The "droplet" mode of the particle number size distribution (PNSD). Factor loadings calculated by PNSD Principal Component Aanalysis (PCA) are plotted against electrical mobility particle diameter ( $d_m$ ). Factor loadings are PCA statistical variables indicating correlations between PNSDs and the droplet mode PCA component. Data from this study (brown line) are compared with data

930 obtained by Costabile et al. (2009) in Leipzig (Germany) at: (a) combined urban sites for 70 days in Spring 2005 (green, blue, red, and black lines, as indicated in the legend), and (b) a single site from long-term (2 years) measurements (black line).

---

## Author Response (AR3)

**LETTER TO EDITOR**

Dear Editor,

thank you very much for your kind handling of our manuscript. We would also like to thank the referees for the constructive comments. We have carefully revised the manuscript according to your suggestions, and are herewith sending the revised version of the manuscript, including a point-by-point reply to the comments.

We hope that the revised manuscript will now be considered appropriate for publication.
Sincerely yours,
Francesca Costabile (on behalf of all of the co-authors)

**Point-by-point reply to the comments**

All referee comments (RC) are in bold, our answers (AA) in normal font, and changes in the manuscript are kept in Italics.

**RC. "First, the finding that there is a significant dependence of BrC on the BC-to-OA ratio (e.g. line 360-362) does not necessarily follow from the data. The dependence of AAE on BC-to-OA ratio shown in Figure 3 can be due to the mere fact that at low BC-to-OA ratios OA has more weight in dictating AAE."**

**"I believe that ... the parameterization of AAE as a function of BC-to-OA ratio is useful, but it should be presented as an average property (for both BC and OA), and not an indication of the optical properties of BrC."**

AA. We agree with the referee. We revised the manuscript (lines 363-373) as follows:

*The increase of the AAE with the decrease of the BC-to-OA ratios is illustrated in Figure 3. The relation observed between AAE and BC-to-OA is an average aerosol property (i.e., for both BC and OA) as suggested by the parametrization of AAE as a function of the BC-to-OA ratio (inner panel of Fig.3) consistent with Lu et al. (2015). The grey line (referring to all the data points) shows the parametrization for the bulk aerosol. The black line (referring to the subset of all the data with PC3 scores > 0) shows the parametrization for BrC. We note a larger dependence of AAE on the BC-to-OA ratio for BrC (r=-0.86, p<0.001) - as opposed to the bulk aerosol (r=-0.7, p<0.001). However, this can be due to the fact that at low BC-to-OA ratios OA has more weight in dictating AAE.*

**RC. "Second, the concern that I raised in the last two reviews about AAE plateauing at 1.8 for large BC-to-OA ratios still holds."**

AA. We thank the referee for his/her observations. In the revised manuscript, the authors acknowledge that (lines 375-389):

*BC and AAE derived from PSAP measurements are affected by uncertainty related to the assumption employed to convert raw PSAP measurements to absorption coefficients, which is about ± 40% based on literature studies (Lack et al, 2008; Nakayama et al., 2010; Lack and Langridge, 2013; Bond et al., 2013; Backman et al., 2014). This uncertainty could in principle reconcile our plateau AAE values higher than 1 with the common assumption that AAE tends to 1 when the aerosol is dominated by BC. Nevertheless, we modeled AAE as a function of aerosol chemical and microphysical properties (Fig. 1). Such simulation clearly shows that AAE close to 1 can be obtained for aerosol population dominated by fresh fossil fuel combustion emissions, with diameter centered below 20 nm. Conversely the size distribution of aerosol population investigated in this study is centered around 80-300 nm (Fig. 1). In addition source apportionment studies performed in the Po valley show that carbonaceous aerosol, both in urban and rural sites, is strongly affected by wood burning emissions (Gilardoni 2016, Gilardoni 2011, Larsen 2012). It follows that, even considering the possible bias due to the experimental uncertainty, the AAE plateauing at values higher than 1 for large BC-to-OA ratios is still is in agreement with model results and source apportionment data.*

**RC. "please clearly explain in more detail how this manuscript relates to the closely related paper from your group by Gilardoni et al. that was published in PNAS recently. The two papers clearly relate strongly to each other. This overlap and similarity is not necessarily a negative, provided that the two papers add to each other and do not repeat an excessive amount of information and results, and provided that the relationship between the two papers is presented clearly."**

AA. We thank the referee for his/her comment, which was addressed in the revised manuscript (lines 490-497) as follows:

*In the paper by Gilardoni et al. (2016) we investigated organic aerosol source and identified SOA formation from processing of biomass burning emissions in the aqueous phase. We than discussed the climate implication of this aqSOA formation at the light of its optical properties, including AAE. In the present paper we investigated the particle size distribution and spectral optical properties of brown carbon in the ambient aerosol, and related these to major aerosol chemical components. By combining the analysis of mycrophisical and optical properties reported here with the source apportionment study by Gilardoni et al. (2016), we demonstrated that the aqSOA formation adds mass to the atmospheric aerosol in the droplet mode range.*

**RC. The main aspect that is missing is the authors do not show how AAE depends on ammonia, other than a high correlation in Table 1. Since ammonia has been identified as a key reactant in brown carbon formation (1-8), this relationship is of special interest to atmospheric chemists, and the papers impact would be greatly increased by including this data analysis, e.g. in Figure 2.**

AA. We would like to thank the referee for this suggestion. We investigated the dependency of AAE on ammonium concentration in the updated Figure 2 (unfortunately gas phase ammonia concentration was not available during the experiment). At the light of this figure, in the revised manuscript we conclude that *BrC occurs in particles ... enriched in ammonium nitrate, and poor in BC* (lines 6-7, 353, 391-392, 487-489).

Relevant references [1-8] were added (lines 48-49).

**Line 80: How/why is dust associated with high fOA? Or is this only an attribute of HULIS?**

AA. We agree with the referee that this sentence is not clear enough. In the revised manuscript we keep the indication that *on the basis of the same data, Shinozuka et al. (2009) showed that AAE generally increases as $f_{OA}$ or SSA increases.*

**Line 453: Somewhere in the manuscript the authors should explain why these index of refraction values are peculiar**

AA. The referee is right, and we apologize. We explained in the revised manuscript (lines 341-343) that *these are $\lambda$ dependent values consistent with BrC in the ambient atmosphere (Moise et al., 2015) in an air mass having high OC to sulfate ratio and nitrate to sulfate ratio (Flowers et al., 2010).* Also, we believe that the term "peculiar" may be confusing and we used "specific".

**RC. Figure 5: Two of these data sets (a and c) have been shown in previous figures on the same axes, but with different variables used for color coding. Given that the color coding looks about the same, this suggests that a graph of albedo vs droplet mode aerosol score might be useful. Just a suggestion.**

AA. In the revised manuscript we added the graph of albedo vs droplet mode aerosol score in Figure 4 of the supplementary material (introduced at lines 355).

**Technical comments**
**Line 29: This statement, while correct, should be accompanied by several literature references.**

AA. We referred to the literature reviews by Laskin et al (2015) and Moise et al (2015).
**Line 43: This statement is inaccurate unless qualified. Are the authors referring only to field measurement studies?**

AA. Yes, and we corrected the text accordingly.
**Line 94: The abbreviation PSAP is used here before it is defined 3 paragraphs later.**

AA. The abbreviation PSAP is now defined here.

**Line 166: Please include the 50% cut-off diameter for the impactor at the SMPS flow rates used.**

AA. We indicated the 50% cutoff diameter = 0.457 $\mu$m at the aerosol flowrate = 0.6 lpm.

**Line 384: The reference to Figure 1 in the supplementary material is confusing. The figure caption makes no mention of showing case study data, and the figure does not identify the case study data in any way.**

AA. We properly indicated the case study day in the caption of Figure 1 in the supplementary material.

**Line 399: mycrophisical should be microphysical**

AA. Done.

**Figure 1: the PC2 symbol in the legend above the graph is not the same as the one on the graph**

AA. Done

**Figure 4: Is the local time at the Po Valley sampling site the same at UTC? There should be a statement to this effect.**

AA. The revised caption of Fig.4 indicates that *Local Time at the Po Valley sampling site in this period is UTC + 1 hour.*

**Also note that both Referees rated the presentation quality of this manuscript as only "Fair". This reflects the often clunky and unclear language used. Please carefully review your writing and rephrase to increase the clarity of the writing as needed.**

AA. We agree with this comment, and we do apologize. We rephrased the revised manuscript to increase the clarity.

[revised manuscript text omitted]